**Inferring the effects of sink strength on plant carbon balance processes from**
**experimental measurements**
Kashif Mahmud[1], Belinda E. Medlyn[1], Remko A. Duursma[1], Courtney Campany[1,2], Martin
G. De Kauwe[3]
[1]Hawkesbury Institute for the Environment, Western Sydney University, Locked Bag 1797,
Penrith NSW 2751, Australia
[2]Department of Biology, Colgate University, NY 13346, USA
[3]ARC Centre of Excellence for Climate Extremes, University of New South Wales, Sydney,
NSW 2052, Australia
*Correspondence to*: Kashif Mahmud (k.mahmud@westernsydney.edu.au)
**Abstract**
The lack of correlation between photosynthesis and plant growth under sink-limited
conditions is a long-standing puzzle in plant ecophysiology that currently severely
compromises our models of vegetation responses to global change. To address this puzzle,
we applied data assimilation to an experiment where sink strength of *Eucalyptus tereticornis*
seedlings were manipulated by restricting root volume. Our goals were to infer which
processes were affected by sink limitation, and to attribute the overall reduction in growth
observed in the experiment, to the effects on various carbon (C) component processes. Our
analysis was able to infer that, in addition to a reduction in photosynthetic rates, sink
limitation reduced the rate of utilization of non-structural carbohydrate (NSC), enhanced
respiratory losses, modified C allocation and increased foliage turnover. Each of these effects
was found to have a significant impact on final plant biomass accumulation. We also found
that inclusion of a NSC storage pool was necessary to capture seedling growth over time,
particularly for sink limited seedlings. Our approach of applying data assimilation to infer C
balance processes in a manipulative experiment enabled us to extract new information on the
timing, magnitude, and direction of the internal C fluxes from an existing dataset. We suggest
this approach could, if used more widely, be an invaluable tool to develop appropriate
representations of sink-limited growth in terrestrial biosphere models.
**Keywords:** Non-structural carbohydrate, carbon allocation, data assimilation, mass-balance,
photosynthesis, plant growth, sink regulation

## 1  Introduction

Almost all mechanistic models of terrestrial vegetation function are based on the carbon (C)
balance: plant growth is represented as the difference between C uptake (through
photosynthesis) and C loss (through respiration and turnover of plant parts). This approach to
modeling plant growth dates back to early crop and forest production models (McMurtrie and
Wolf, 1983; de Wit and van Keulen, 1987; de Wit, 1978) and now provides the fundamental
quantitative framework to integrate our scientific understanding of plant ecosystem function
(Makela et al., 2000).
However, C balance models have been criticized for being "source-focused" (Fatichi et al.,
2014). Most C balance models predict growth from the environmental responses of
photosynthesis ("source limitation"). In contrast to this assumption, many experimental
studies demonstrate that growth is directly limited by environmental conditions ("sink
limitation") rather than the availability of photosynthate. For example, growth is more
sensitive to water limitation than is photosynthesis (Bradford and Hsiao, 1982; Müller et al.,
2011; Mitchell et al., 2014); low temperatures are considerably more limiting to cell division
than to photosynthesis (Körner et al., 2014); nutrient limitation may slow growth without
reducing photosynthesis (Reich, 2012; Crous and Ellsworth, 2004); and, physical sink-
limitation may reduce growth with a decline in photosynthetic capacity and an accumulation
of leaf starch (Arp, 1991; Campany et al., 2017; Poorter et al., 2012a; Paul and Foyer, 2001).
How can we move to models that include both source- and sink-limitation? There is ongoing
discussion about realistic implementations of non-structural carbohydrates (NSC) in
vegetation models because of their multiple roles in plant functioning, such an
implementation provides a buffer against discrepancies in source and sink activity. Some C
balance models include a "storage" pool of NSC (Running and Gower, 1991; Bossel, 1996;
Thornley and Cannell, 2000), but most of these models make the assumption that the NSC
pool acts merely as a buffer between C sources and sinks, balancing out seasonally or at least
over several seasons (Fatichi et al., 2014; Friend et al., 2014; De Kauwe et al., 2014; Schiestl-
Aalto et al., 2015). There is mounting evidence that the NSC plays a more active role in tree
physiology (Buckley, 2005; Sala et al., 2012; Wiley and Helliker, 2012; Hartmann et al.,
2015). For example, NSC accumulation can lead to down-regulation of photosynthesis
(Nikinmaa et al., 2014). Therefore, the need to quantify the NSC pool and to better
understand the prioritisation of storage vs. growth is of great importance.
An understanding of the dynamics of storage is also essential to correctly represent the C
balance in models (Hartmann and Trumbore, 2016). If, for example, a direct growth
limitation is implemented into models, how should the surplus of accumulated photosynthates
be treated? In their proof-of-concept sink-limited model, Fatichi et al. (2014) allowed
reserves to accumulate indefinitely. Alternatively, some models (e.g. CABLE (Law et al.,
2006), O-CN (Zaehle and Friend, 2010)) increase respiration rates when excess labile C
accumulates. Both approaches can be seen as model-oriented solutions to maintain C balance
that are unsatisfactory because they are not based on empirical data. Experiments where sink
strength is manipulated may provide the key to improve our understanding of C balance
processes under direct growth limitation.
Efforts have been made to understand the physiological and morphological changes in
response to belowground C sink limitation by manipulating rooting volume in tree seedlings
(Arp, 1991; Campany et al., 2017; Poorter et al., 2012a). These experiments often reveal
photosynthetic down-regulation and accumulation of leaf starch, and reductions in growth
(Arp, 1991; McConnaughay and Bazzaz, 1991; Gunderson and Wullschleger, 1994; Sage,
1994; Poorter et al., 2012a; Robbins and Pharr, 1988; Maina et al., 2002; Campany et al.,
2017). In a recent study with Eucalyptus seedlings, Campany et al. (2017) showed that the
reduction in seedling growth when rooting volume was restricted could not be completely
explained by the negative effects of sink limitation on photosynthesis, suggesting that other
components of the C balance were affected in the process. However, Campany et al. (2017)
could not accurately quantify all components of tree C balance, i.e. photosynthesis,
carbohydrate storage, biomass partitioning and respiration.
Quantifying all components of C balance is not an easy task, given that not all processes are
measured with equal fidelity, and data gaps will always occur. Klein and Hoch (2015) used a
C mass balance approach with a tabular process flowchart to decipher C components and
provide a full description of tree C allocation dynamics. Here, rather than using a manual
process, we used a data assimilation (DA)-modelling framework, which has been proven to
be a powerful tool in analyzing complex C balance problems (Williams et al., 2005;
Richardson et al., 2013). For example, Richardson et al. (2013) use DA to discriminate
among alternative models for the dynamics of non-structural carbon (NSC), finding that a
model with two NSC pools, fast and slow, performed best; Rowland et al. (2014) applied DA
to experimental observations of ecosystem C stocks and fluxes to infer seasonal shifts in C
allocation and plant respiration in an Amazon forest; and Bloom et al. (2016) used DA to
constrain a C balance model with satellite-derived measurements of leaf C, to simulate
continental-scale patterns in C cycle processes.
Our goal in this paper was to use DA to quantify the impact of sink limitation on C balance
processes. We utilized data from an experiment in which sink limitation was induced by
restricting the rooting volume of *Eucalyptus tereticornis* seedlings over the course of 4
months (Campany et al., 2017). We assimilated photosynthesis and growth measurements
from the experiment into a simple C balance model, to infer the effects of sink limitation on
the main C balance processes, namely: respiration, carbohydrate utilization, allocation, and
turnover.
Although in reality plants do have a storage component, it is not necessarily the case that
including such a storage component in the model leads to model improvement. Hence, it is
important to test whether or not adding the storage component improves the performance of
the model enough to justify the additional complexity. Therefore, we first tested two null
hypotheses:
H1: There is no need to consider storage in the model: growth can be adequately predicted
from current day photosynthate.
H2: There is no effect of sink limitation on C balance processes other than via a reduction of
photosynthesis.
We were then interested to test the following specific hypotheses about the impact of sink
limitation on C balance:
H3: We hypothesized that the rate of utilization of carbohydrate for plant growth would be
lower under sink limitation, causing growth rates to slow and non-structural carbohydrate to
accumulate.
H4: We hypothesized that under sink limitation a larger proportion of C would be lost to
growth respiration and less used for production. We have dubbed this the "wasteful plant"
hypothesis; this hypothesis corresponds to the assumption embedded in some models that
respiration is up-regulated when labile C accumulates e.g. CABLE, O-CN (Law et al., 2006;
Zaehle and Friend, 2010).
H5: We hypothesized that foliage and root C allocation fractions would be reduced, in favour
of wood allocation. Sink limitation induced by nutrient and/or water stress often results in a
shift in C allocation away from foliage and towards fine roots (Poorter et al., 2012b).
However, for this experiment, the physical restriction of root growth limits the potential for
root allocation. Hence, we predicted that both foliage and fine root allocation would decrease.

**2    Materials and Methods**
**2.1    Experiment description**
The site and experimental setup have been described in detail by Campany et al. (2017), so
we only provide a brief description here. The experiment was carried out at the Hawkesbury
Forest Experiment site (33°37'S 150°44'E) in Richmond, NSW, Australia. The site is located
in the sub-humid temperate region and experiences warm summers and cool winters. The
seedlings were planted on 21$^{st}$ January 2013 (mid-summer) and harvested on 21$^{st}$ May 2013
(late autumn).  Mean daily temperatures ranged from 22.8 to 46.4 °C (monthly mean of 32.1
°C) in January 2013, which was the warmest month of the year, and cooled down in May
2013 with an average of 21 °C  (BoM, 2017).
Twenty-week old *Eucalyptus tereticornis* seedlings in tube stock were chosen from a single
local Cumberland plain cohort. Ten seedlings were harvested at the start of the experiment to
measure initial leaf area and dry mass of foliage, woody components and roots. Forty-nine
seedlings were used in the main experiment, allocated to seven treatments. The plants were
grown in containers of differing volume set into the ground (5, 10, 15, 20, 25 or 35 L), or
were planted directly into soil (free seedlings, used as the control). All plants were grown in
the open under field conditions, but were watered regularly to avoid moisture stress.

### 2.2  Experimental data acquisition

Full details of all measurements are given in Campany et al. (2017). The mass of each pool
(foliage, wood, root, storage) was estimated over time as follows. The initial dry mass of
leaves, woods and roots was measured for 10 seedlings at the start of the experiment using
the harvesting procedure described in Campany et al. (2017). The dry mass of all
experimental plants was measured at the end of the experiment following the same procedure.
Seedling growth was tracked during the four months of the experiment, by measuring stem
height (h), diameter at 15 cm height (d) and number of leaves on a weekly basis. These
measurements were used to estimate the time course of wood and foliage biomass: for root
total C we used only initial and final harvest measurements. Initial root C was estimated by
averaging all 10 harvested seedlings.
We estimated weekly total C in wood ($C_{s,w}$) from the measurements of stem height and
diameter, by using an allometric model fitted to initial and final harvest data.

$$\log(C_{t,w}) = b_1 + b_2 \log(d) + b_3 \log(h) \tag{1}$$

For each seedling, the total leaf area (LA) and foliage total C ($C_{t,f}$) over time (t) were
estimated based on harvested data (T = time of harvest) and weekly leaf counts (LC) over
time.

$$LA\ (t) = \frac{LA\ (T)}{LC\ (T)}\ LC\ (t) \tag{2}$$

$$C_{t,f}\ (t) = \frac{M_f\ (T)}{LC\ (T)}\ LC\ (t) \tag{3}$$

Fully expanded new leaves were sampled for total non-structural carbohydrate (NSC)
concentration on a fortnightly basis. These concentrations were multiplied by leaf biomass to
estimate the foliage TNC pool ($C_{n,f}$) at each time point. The partitioning of the non-structural
C amongst foliage, wood and root tissues, according to empirically-determined fractions, was
then used to estimate the wood and root components of the total TNC pool. Structural C mass
for each component was estimated by subtracting non-structural C mass from total C mass.
Only foliage non-structural C ($C_{n,f}$) was measured, so to estimate the partitioning of the non-
structural C among different organs, we used data from a different experiment on similar-
sized seedlings of a related species (*Eucalyptus globulus*), which were grown in 5L pots until
four months of age (Duan et al., 2013). We used data from the ambient well-watered control
treatments. In that experiment, foliage, wood and root NSC were measured repeatedly over
two months. There was no statistically significant change over time in the NSC distribution,
so we used the mean distribution for mass-specific $C_n$ over time, which was calculated to be a
ratio of 75:16:9 among foliage, wood and root pools.
We estimated daily GPP from leaf-level gas exchange measurements and a simple canopy
scaling scheme as described in Campany et al. (2017), and summarized below. Measurements
of photosynthesis were made fortnightly throughout the experiment on one fully expanded
leaf per plant (Campany et al., 2017). Photosynthetic $CO_2$ response (ACi) curves and leaf
dark respiration rates (R) were measured on two occasions, 13-14[th] March 2013 (when new
leaves were first produced) and 14-15[th] May 2013 (prior to the final harvest). The ACi curves
were used to estimate photosynthetic parameters (the maximum rate of Rubisco
carboxylation, $V_{cmax}$ and the maximum rate of electron transport for RuBP regeneration under
saturating light, $J_{max}$) using the biochemical model of Farquhar et al. (1980) and fit with the
'plantecophys' package (Duursma, 2015) in R. The parameter $g_1$, reflecting the sensitivity of
stomatal conductance ($g_s$) to the photosynthetic rate, was estimated by fitting the optimal
stomatal conductance model of Medlyn et al. (2011) to measured stomatal conductance data.
Treatment effects on photosynthesis were detected immediately on newly produced (fully
expanded) leaves and Campany et al. (2017) did not observe variation over time in
photosynthetic rates. Hence, the photosynthesis parameters were assumed not to change over
time but were specific for each treatment. Therefore, daily net C assimilation per unit leaf
area ($C_{day}$) was estimated by using a coupled photosynthesis–stomatal conductance model
(Farquhar et al., 1980; Medlyn et al., 2011) using mean photosynthetic parameters ($J_{max}$,
$V_{cmax}$, $g_1$ and $R_d$) for each treatment and meteorological data from the onsite weather station.
The daily GPP was estimated by multiplying $C_{day}$, total leaf area (LA) and a self-shading
factor. The self-shading factor, which is a linear function of LA, is calculated by via
simulation with a detailed radiative transfer model, the 'YplantQMC' R package of Duursma
(2014) for individual treatment. The leaf maintenance respiration rate ($R_m$, g C g$^{-1}$ C plant d$^{-1}$)
was calculated for each seedling by scaling the measured rate (R) to air temperature using a
$Q_{10}$ value of 1.86 (Campany et al., 2017). The daily total maintenance respiration, $R_{m,tot}$ is
calculated as a temperature-dependent respiration rate, $R_m$, multiplied by plant biomass. We
assumed the same tissue-specific dark respiration rates for leaf, woody and root tissues for
these seedlings, as was observed for seedlings of this species by Drake et al. (2017).

## 210     **2.3    Carbon Balance Model (CBM)**

We used a DA-modelling framework, similar to that used by Richardson et al. (2013). This
approach uses a simple carbon balance model shown in Figure 1. The model is driven by
daily input of gross primary production (GPP), which directly enters into a non-structural C
pool ($C_n$). The daily total maintenance respiration, $R_{m,tot}$, is subtracted from $C_n$ pool. The pool
is then utilized for growth at a rate $k$ (i.e. $kC_n$). Of the utilization flux, a fraction $Y$ is used in
growth respiration ($R_g$), and the remaining fraction ($1-Y$) is allocated to structural C pools
($C_s$): among foliage, wood and root ($C_{s,f}$, $C_{s,w}$, $C_{s,r}$). The foliage pool is assumed to turn over
with rate $s_f$. We assume there is neither wood or root turnover as the seedlings in the
experiment were young.

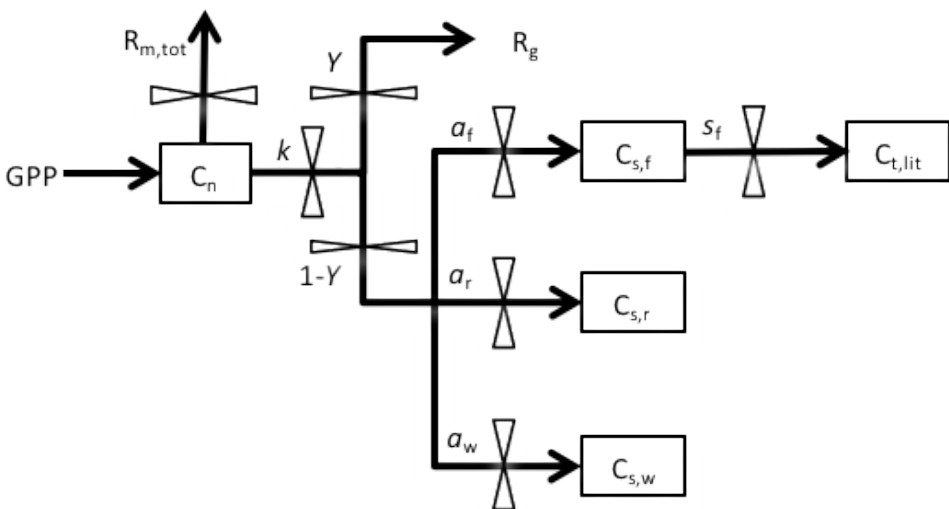


**Figure 1:** Structure of the Carbon Balance Model. Pools, shown as boxes: $C_n$, non-structural
storage C; $C_{s,f}$, structural C in foliage; $C_{s,r}$, structural C in roots; $C_{s,w}$, structural C in wood.
Fluxes, denoted by arrows, include: GPP, gross primary production; $R_{m,tot}$, total maintenance
respiration; $R_g$, growth respiration; $C_{t,lit}$, structural C in leaf litterfall. Fluxes are governed by
six key parameters: k, storage utilization coefficient; Y, growth respiration fraction; $a_f$,
allocation to foliage; $a_w$, allocation to wood; $a_r$, allocation to roots; $s_f$, leaf turnover rate. $a_r$ is
defined as $1 - a_f - a_w$.

The dynamics of the four carbon pools are described by four difference equations:

$$\Delta C_n = GPP - R_m(C_{t,f} + C_{t,w} + C_{t,r}) - k\,C_n \tag{4}$$

$$\Delta C_{s,f} = k\,C_n\,(1-Y)\,a_f - s_f\,C_{s,f} \tag{5}$$

$$\Delta C_{s,w} = k\,C_n\,(1-Y)\,a_w \tag{6}$$

$$\Delta C_{s,r} = k\,C_n\,(1-Y)\,a_r \tag{7}$$

Where GPP is the gross primary production (g C plant$^{-1}$ d$^{-1}$); $R_m$ is the maintenance respiration rate (g C g$^{-1}$ C d$^{-1}$); $C_{t,f}$, $C_{t,w}$, and $C_{t,r}$ are the total C in foliage, wood and root respectively (g C plant$^{-1}$); $k$ is the storage utilization coefficient (g C g$^{-1}$ C d$^{-1}$); $Y$ is the growth respiration fraction; $a_f$, $a_w$, $a_r$ are the allocation to foliage, wood and root respectively; and $s_f$ is the leaf turnover rate (g C g$^{-1}$ C d$^{-1}$). $a_r$ is defined as $1 - a_f - a_w$.

The non-structural (storage) C pool ($C_n$) is assumed to be divided amongst foliage, wood and root tissues ($C_{n,f}$, $C_{n,w}$, $C_{n,r}$) according to an empirically-determined ratio of 75:16:9. Total carbon in each tissue ($C_t$) is then calculated as the sum of non-structural carbon ($C_n$) and structural carbon ($C_s$) for that tissue.

$$C_{t,f} = 0.75 \times C_n + C_{s,f} \tag{8}$$

$$C_{t,w} = 0.16 \times C_n + C_{s,w} \tag{9}$$

$$C_{t,r} = 0.09 \times C_n + C_{s,r} \tag{10}$$

## 2.4 Application of Data Assimilation (DA) algorithm

DA was used to estimate the six parameters ($k$, $Y$, $a_f$, $a_w$, $a_r$, $s_f$) of the CBM for this experiment. All parameters were allowed to vary quadratically with time, i.e. each parameter was represented as:

$$p = p_1 + p_2 t + p_3 t^2 \tag{4}$$

Quadratic variation over time was found to yield significantly better model fits than either constant parameter values or linear variation over time (see supplementary section S1). We executed three distinct sets of model simulations (Table 1), with the goals of (1) testing the need for a storage pool; (2) determining the effect of sink limitation on model parameters; and (3) attributing the overall effect of sink limitation on growth to the change in individual parameters.

For each set of model simulations, GPP and $R_m$ were used as inputs to the DA framework, and the measurements of total C mass of each of the plant components and foliage NSC concentrations were used to constrain the parameter values. The set of constraints included 18 measurements of $C_{t,f}$ and $C_{t,w}$, two measurements of $C_{t,r}$ (start and end of the experiment), and six measurements of foliage NSC. There were 5 quadratically-varying parameters to determine for each treatment, summing to a total of 15 (3x5) coefficients to determine, compared with total 44 data measurements available, for each treatment.

We used the Metropolis algorithm (Metropolis et al., 1953) as implemented by Zobitz et al. (2011), with broad prior Probability Density Functions (PDFs) for the parameters (Table 2). Values of $k$, $a_f$, $a_r$ and $s_f$ were allowed to vary within the maximum possible range, while parameter $Y$ was constrained according to the literature on growth respiration (Villar and Merino, 2001) . Parameter $a_r$ was calculated from $a_f$ and $a_w$ with a check on $a_r$ to ensure that it had reasonable values ($0 < a_r < 1$). Standard Error (SE) was used as an estimate of uncertainty on the assimilated data (Rowland et al., 2014; Richardson et al., 2010), and was calculated based on six replicate measurements. When combining errors, the errors were assumed to be uncorrelated (Hughes and Hase, 2010).

Model parameters were assumed to be real, positive and to have a lognormal probability distribution (Rowland et al., 2014). Therefore, all processes of parameter selection, and acceptance and rejection of parameters in relation to prior ranges were performed in lognormal space (Knorr and Kattge, 2005). We performed the first iteration starting from the prior set of parameters. To generate subsequent values for each parameter, a new point was generated by varying all vector elements by some step, chosen with a Gaussian distributed random number generator having a mean of 0 and a SD of 0.005 in log-normal space. We adjusted the step length for each parameter to values which lead to an average acceptance rate of the new points around 35–40%. We confirmed the chain convergence, having 3000 iterations to adequately explore the posterior parameter space, by visual inspection of the trace plots of different parameters as suggested by Van Oijen (2008). The trace plots show how the chain moves through parameter space for each individual parameter. The parameter vectors sampled during the first phase of the chain were not representative and therefore the first 10% of the chain was discarded from the posterior sample.

**Table 1:** Summary of the three model simulation sets

| Simulation Set | Goal | Features | Addressing hypothesis |
|---|---|---|---|
| A | Test importance of storage pool | • DA applied to estimate parameters for model without storage pool and model with storage pool<br>• Three treatment groups<br>• Not constrained with NSC data<br>• No leaf area feedback | H1 |
| B | Identify effect of sink limitation on model parameters | • DA applied to estimate parameters for model with storage pool<br>• Data divided into one, two, three or seven treatment groups<br>• Constrained with NSC data<br>• No leaf area feedback | H2-H5 |
| C | Attribute overall effect on growth to changes in individual parameters | • Forward model runs to quantify impact of individual processes on overall plant growth<br>• 5L and free seedlings treatments considered<br>• Parameters changed individually and sequentially<br>• Leaf area feedback on photosynthesis and $R_m$ | |

279    **Table 2:** Prior parameter PDFs (with uniform distribution) and the starting point of the
280    iteration for all parameters

| Parameter | Minimum | Maximum | Starting value |
|---|---|---|---|
| $k$ | 0 | 1 | 0.5 |
| $Y$ | 0.2 | 0.4 | 0.3 |
| $a_f$ | 0 | 1 | 0.5 |
| $a_w$ | 0 | 1 | 0.5 |
| $s_f$ | 0 | 0.01 | 0.005 |
| $a_r = 1 - (a_f + a_w)$, where $0 < a_r < 1$ | | | |

### 2.4.1 Importance of storage pool

We tested the hypothesis (H1) on the importance of including a non-structural C storage pool in CBM by contrasting fits of the full model with fits of a simplified model without the non-structural C pool (Simulation Set A, Table 1). The simplified model omits the non-structural C pool ($C_s$) from the full model (Figure 1) and assumes that all available C is utilized for growth each day. We applied the DA framework to both model options and calculated the Bayesian Information Criterion, BIC (Schwarz, 1978) to determine the better model structure. BIC measures how well the model predicts the data based on a likelihood function and compare model performance taking into account the number of fitted parameters, with the lowest BIC number indicating the best model setting. For this comparison, both models were fit to the biomass data only, not leaf NSC data, in order to ensure that both models were fit to the same number of data points.

### 2.4.2 Effects of sink limitation on model parameters

The effects of sink limitation on C balance were investigated by applying the DA framework to data from all treatments combined, and then subsets of treatments (Simulation Set B, Table 1). Considering all treatments pooled together gives same parameters for all the treatments and effectively assumes no effect of sink limitation. On the other hand, taking more subsets of treatments produces more parameter sets (one for each subset) and allows for parameters to vary according to the degree of sink limitation. We first fitted the model to all data, ignoring treatment differences; then considered 2 treatment groups (free seedling / 5-35 L containerized seedlings), 3 groups (free / 5–15 L / 20–35 L) and 4 groups (free / 5-10 L / 15-20 L / 25-35 L). We also fitted the model to each of the 7 treatments individually, where the parameter set for each treatment is unique. The BIC values were compared across treatment groupings.

### 2.4.3 Attribution analysis

We performed a sensitivity analysis to quantify the impact of the response of each individual process to sink limitation on overall plant growth (Simulation Set C, Table 1). This analysis consisted of forward runs of the model, including a leaf area feedback to GPP. That is, rather than taking GPP based on measured LA (Eq. 9) as input, in this version of the model we calculated daily GPP using the modelled LA, the photosynthesis rate and corresponding self-shading factor. Adding the LA feedback to the model was necessary to quantify how the treatment effect on individual model parameters affects final seedling biomass through its effect on foliage mass, and consequently GPP, over time.

LA in each time step is estimated from NSC-free specific leaf area ($SLA_{nonsc}$) and the predicted foliage structural carbon ($C_{s,f}$) in that time step. $SLA_{nonsc}$ is calculated at harvest discarding the foliage NSC portion and is assumed to be constant for a given treatment throughout the experiment.

$$LA = SLA_{nonsc} \times C_{s,f} \qquad (12)$$

Once the LA feedback was implemented in the CBM, we ran the model with the inputs and modelled parameters from the smallest pot seedling (5 L), then changed the parameters to those for the free seedling sequentially in order to quantify the effect of each parameter on the final seedling biomass. The parameters we considered for this attribution analysis were: daily photosynthetic rate per unit leaf area ($C_{day}$), maintenance respiration rate ($R_m$), C allocation fractions to biomass ($a_f$, $a_w$, $a_r$), growth respiration rate ($Y$), foliage turnover rate ($s_f$) and utilization coefficient ($k$). We additionally carried out a sensitivity analysis in which we varied each parameter from its baseline value separately.

## 3   Results

### 3.1   Importance of storage pool

First, we tested the null hypothesis (H1) that there is no need for a non-structural carbohydrate storage pool in the carbon balance model. We compared BIC values for model structures with and without a storage pool. Table 3 (Simulation Set A) shows the results for model fits with the optimal grouping strategy (three treatment groups). BIC values were consistently lower for the model including the storage pool; the improvement in model fit is most noticeable for the containerized seedlings. This analysis demonstrates that the model does need to include a storage pool to correctly represent the experimental data. In all remaining analyses, the full CBM (with non-structural C pool) is applied to data from all four plant C pools (NSC, foliage, wood and root biomass).

**Table 3:** BIC values from model fits. The lowest BIC values indicate the best performing parameter settings for any particular simulation. Note that, for Sim A, leaf NSC data were not used to constrain either model, to ensure that both models were fit to the same dataset, resulting in lower BICs compared to Sim B. Treatment groups are: 'Small' - 5 L, 10 L and 15 L containers; 'Large' - 20 L, 25 L and 25 L containers; 'Free' – freely rooted seedlings; 'All' - all data; 'Containerized' - all plants in containers.

| Simulation Set | Model Setting | Treatment groups | BIC |
|---|---|---|---|
| Sim A | Model without storage pool | Small | 459 |
|  |  | Large | 550 |
|  |  | Free | 182 |
|  | Model with storage pool | Small | 215 |
|  |  | Large | 338 |
|  |  | Free | 167 |
| Sim B | 7 treatments combined | All | 2768 |
|  | 2 groups | Containerized | 1813 |
|  |  | Free | 170 |
|  |  | Total | 1983 |
|  | 3 groups | Small | 683 |
|  |  | Large | 457 |
|  |  | free | 170 |
|  |  | Total | 1310 |
|  | 7 treatments individually | 5 L | 85 |

| | | 10 L | 98 |
|---|---|---|---|
| | | 15 L | 60 |
| | | 20 L | 63 |
| | | 25 L | 106 |
| | | 35 L | 152 |
| | | Free | 170 |
| | | Total | 734 |

## 3.2 Sink limitation effect on C balance processes

We addressed our second null hypothesis (H2), that there is no effect of sink limitation on carbon balance processes, by comparing BIC values obtained for model fits when all treatments were combined vs separating the treatments into sub-groups. If there was no effect of sink limitation, the BIC value when all treatments are fit together would be similar to that obtained when treatments are separated into groups. The BIC values shown in Table 3 (Simulation Set B) decrease strongly as number of treatment groups increases, indicating a clear effect of sink limitation on carbon balance processes. Although the BIC values continue to decrease as more treatment groups are considered, we also found that interpreting parameter changes became more difficult as the number of groups increased. Hence, further analyses in this paper used unique parameter sets for three treatment groups: small containers, large containers, and free seedlings.

## 3.3 Analysis of carbon stock dynamics

Figure 2 shows the correspondence between modeled C pools and data. The model reproduced the key features of biomass growth over time in response to treatment. Biomass growth (Figure 2A, B and C) and the foliage storage pool (Figure 2D) were very clearly impacted by sink limitation: biomass growth was strongly reduced for containerised seedlings, which was very well mimicked by the model. Foliage growth in the free seedlings slowed towards the end of the experiment. Wood and root growth continued throughout the experiment in freely-rooted seedlings but slowed down during the second half of the experiment in containerized seedlings. NSC concentrations ($C_{n,f}$ / $C_{t,f}$) in seedlings in small containers were higher compared those in free seedlings at the beginning of the season but all treatments had similar concentrations after four months (Figure 2D). In March, at the time of the first leaf NSC measurements, the foliage storage pool (Supplementary Figure S1) was similar in size across all treatments, but it increased over time in the free seedlings as these plants continued to grow, and decreased over time in the plants in small containers.

Modelled C stocks for all 7 treatments closely tracked their corresponding observations (Figure 2) as most of the predicted biomass values were within one standard error of the measurements. The exception is the 35 L container treatment, which is underestimated slightly because the grouping of 20, 25 and 35 L treatments into one group makes it difficult for the model to fit all treatments in this group.

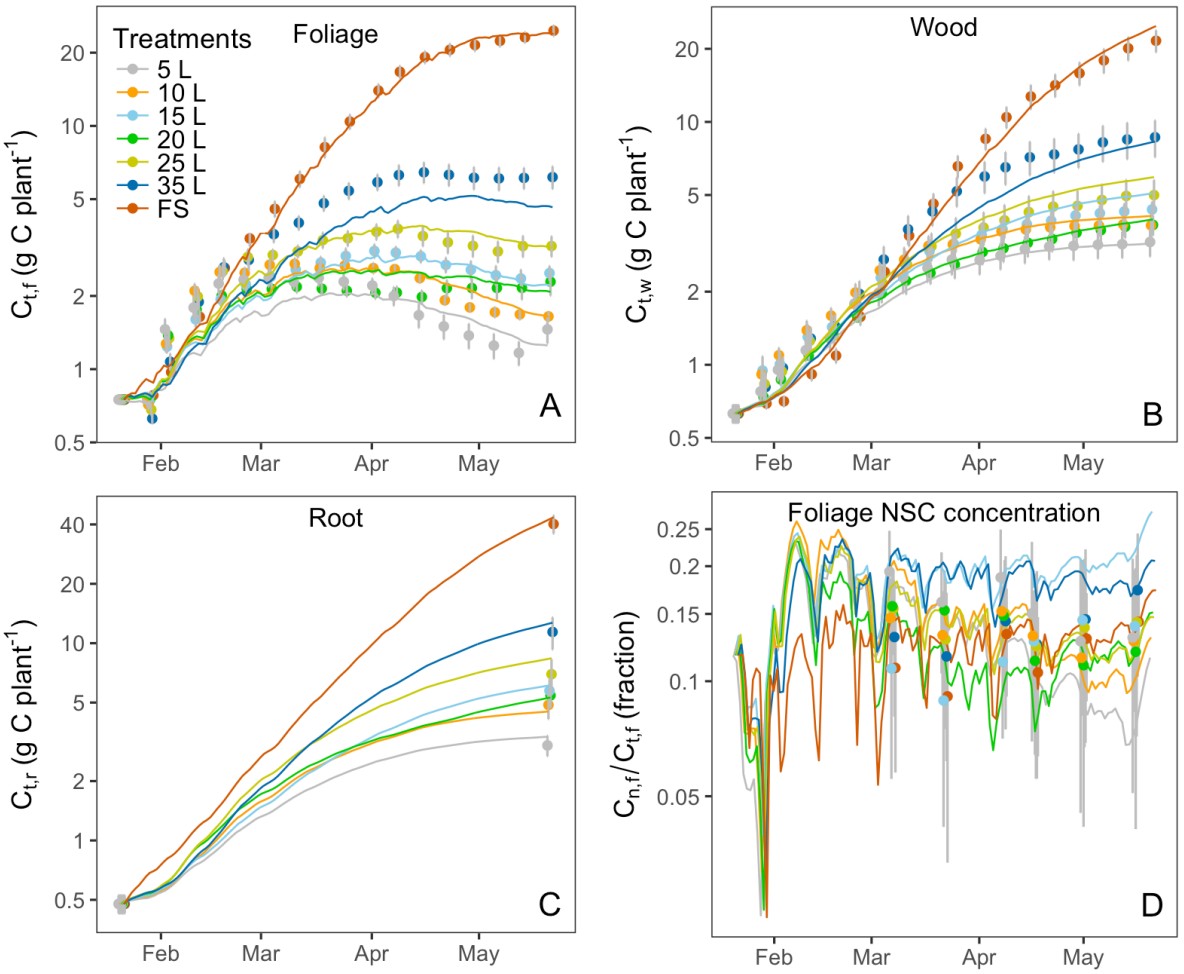

**Figure 2:** C stocks (lines) with the inferred parameter set and corresponding observations (symbols): (A) total C mass in foliage $C_{t,f}$, (B) total C mass in wood $C_{t,w}$, (C) total C mass in root $C_{t,r}$ and (D) foliage NSC concentration ($C_{n,f}/C_{t,f}$). Note that the carbon pools and foliage NSC concentration (y-axes) are plotted on log scale to visualize the changes at the beginning of the experiment. Error bars (1 SE, n = 6) are shown for each observation.

## 3.4 Parameter estimates

Data assimilation indicated significant treatment effects on all five fitted parameters (Figure 3). There was a large effect of sink limitation on the utilization coefficient ($k$). In agreement with our hypothesis H3, the free seedling had the highest $k$, and the seedlings in small containers (most sink limited) had the lowest $k$ (Figure 3A). As the experiment progressed, the utilization rate of free seedlings began to decrease (Figure 3A). In contrast to the free seedlings, the potted seedlings had relatively low utilization rates initially ($k$ close to 0.5) and the utilization rates slowed down abruptly with time, most significantly in the smallest container treatments (Figure 3A).

In agreement with hypothesis H4, the estimated growth respiration rate ($Y$) varied according to the sink strength of the treatment groups, and was highest in the lowest sink strength treatments (Figure 3B). Moreover, $Y$ did not vary significantly over time for the sink limited treatment groups. However, the rate of growth respiration for the free seedling slowed down over time.

The data assimilation process also indicated that the growth allocation fractions vary among treatments and over time. Consistent with hypothesis H5, wood allocation fraction was highest in the smallest container treatments, and lowest in the free seedlings (Figure 3D). For the free seedlings, allocation was initially highest to foliage and roots (Figure 3C-E); over time, the plants reduced allocation to foliage and increased it to wood and roots. In the containerized seedlings, allocation was initially highest to wood and foliage; over time, foliage allocation decreased to almost zero and root allocation increased.

The estimated leaf turnover rate, $s_f$ was also notably higher for sink-limited treatments compared to free seedlings (Figure 3F). The large value of modelled leaf litterfall for sink-limited treatments is consistent with observations during the experiment that containerized seedlings had relatively large leaf litterfall, beyond normal senescence. Estimated $s_f$ increased over time for all treatment groups (most notably in free seedlings), due to a combination of ontogeny, seasonal change, and growth restriction in the sink-limited seedlings.

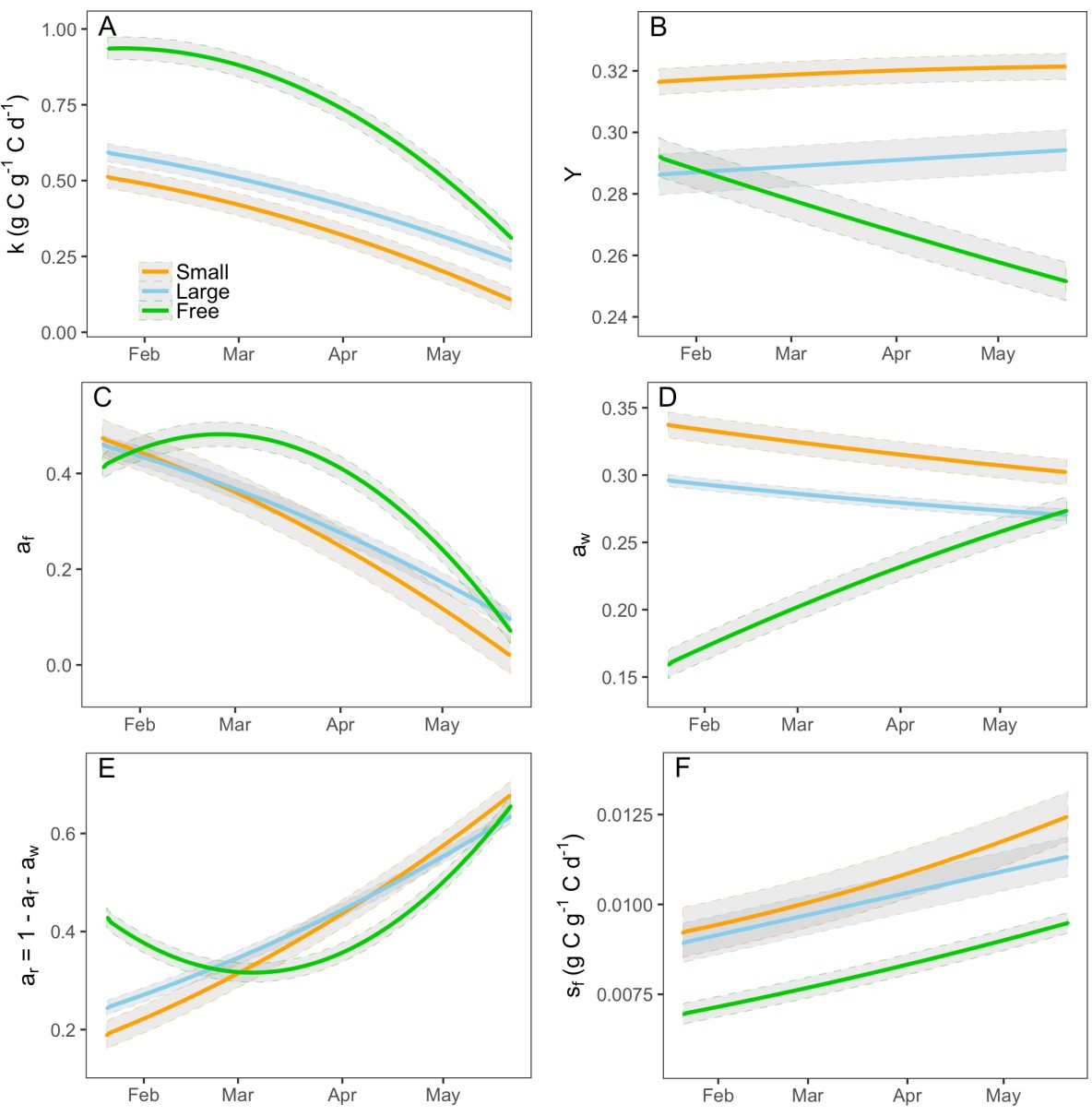

409

**Figure 3**: Modelled final parameters for three groups of treatments during the experiment period (21st Jan to 21st May 2013): (A) storage utilization coefficient, $k$; (B) growth respiration fraction, $Y$; (C) allocation to foliage, $a_f$; (D) allocation to wood, $a_w$; (E) allocation to roots, $a_r$ and (F) leaf turnover rate, $s_f$. $a_r$ is defined as $1 - a_f - a_w$. The grey shaded area shows the 95% confidence intervals of modelled parameters.

**3.5    Carbon budget**

The model was used to partition total GPP (g C plant$^{-1}$) from the entire experiment period into different C pools (growth respiration, maintenance respiration, non-structural carbon, structural foliage, wood, and root carbon, and litterfall) for all 7 treatments (Figure 4). Total

GPP was considerably lower for the containerized seedlings, owing to lower photosynthetic
rates per unit leaf area, $C_{day}$ (Figure 5A), and lower total leaf area (LA) per plant. Though
starting with the same total LA of 0.016 $m^2$, the 5 L containerized and free seedlings had total
LA of 0.031 and 0.516 $m^2$ respectively after four months of treatment. Simultaneously, the
partitioning of GPP changed considerably across different treatments.
Small container seedlings (5, 10, 15 L) had a higher fraction of GPP lost in leaf litterfall
compared to other seedlings (Figure 4), consistent with observations during the experiment.
The proportion of GPP in final foliage mass was extremely low in sink limited treatments
(also shown in Figure 2A). Allocation of GPP to final foliage and root biomass were highest
in the free seedlings, although interestingly allocation to final wood biomass was similar
across treatments. The final allocation to storage was also higher in free seedlings. The sink
limited seedlings had a higher proportional C lost through maintenance respiration. Tissue
specific respiration rates were similar in free and containerized seedlings, so the ~35%
reduction in photosynthetic rate for the smallest containerized seedling, led to a higher overall
$R_{m.tot}$/GPP fraction. In summary, the estimated total respiration ($R_{m.tot} + R_g$) to GPP ratio was
considerably lower for the free seedlings compared to the sink limited treatments. The carbon
use efficiency (CUE) remained relatively constant and high over time for free seedlings
(~0.65), whereas CUE in the smallest container treatments showed a sharp reduction over
time down to ~0.25 (Supplementary Figure S2).

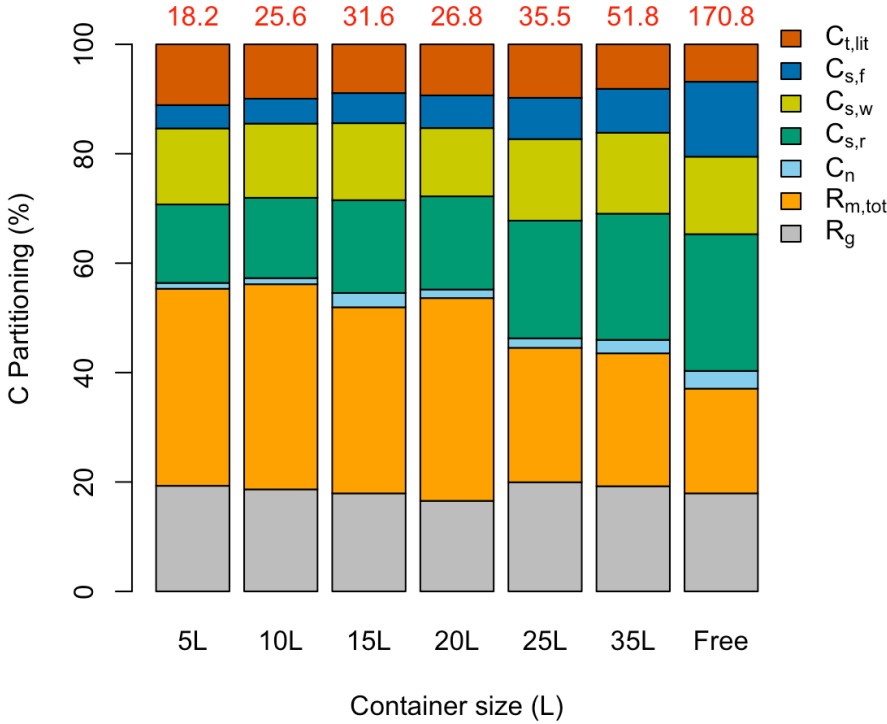

**Figure 4:** Simulated proportional C partitioning for the whole experimental period. The total accumulated GPP (g C plant[-1]) for individual treatments is shown (in red) at the top of each column. Free stands for free seedling. Different C partitions are in the colour legend: total litterfall, $C_{t,lit}$; foliage structural C, $C_{s,f}$, wood structural C, $C_{s,w}$, root structural C, $C_{s,r}$; non-structural C pool, $C_n$; total maintenance respiration, $R_{m,tot}$ and growth respiration, $R_g$.

## 3.6    Attribution analysis

Sink limitation affected biomass growth via a range of processes, namely reduction in photosynthesis, and variation in the utilization rate, growth respiration, leaf litterfall, and C allocations to foliage, wood and root across various treatment groups. We quantified the contribution of each of these process responses separately by running the CBM with parameter inputs changing both sequentially and individually (one at a time). Table 4 presents the effect of the parameters changing individually from the value of the smallest container treatment (5 L) to that of free seedling (FS) and other way around, resetting the previous parameter to the baseline value. The final biomass values in Table 4 indicate the contribution of each individual parameter separately and sequentially. Photosynthetic capacity had the largest individual effect on total plant growth (+15.28 and -71.9 g C) compared to the rest of

the parameters. However, allocation pattern and the utilization rate also had a sizeable effect on final biomass (Table 4).

Figure 5 shows how biomass ($M_f$, $M_w$ and $M_r$) is predicted to change when each parameter is changed sequentially from the parameter set derived from DA on the 5L observations (gray line, Figure 5) to that of the parameters obtained when using the free seedlings as constraint of the model (red line, Figure 5). Daily net C assimilation per unit leaf area ($C_{day}$), which was 30% higher for free seedling compared to 5 L container treatment (Figure 5A), had a large impact on plant growth (final total biomass was increased by 11 g, Table 4 and Figure 5G-I, gray to orange). Maintenance respiration rate ($R_m$) did not vary significantly across treatments (Figure 5B), in line with the data presented in Campany et al. (2017), and consequently its impact was insignificant (the final total biomass is reduced by only 0.24 g, Table 4 and Figure 5G-I, orange to light blue). The modelled biomass allocation fractions ($a_f$, $a_w$ and $a_r$) in Figure 5C had important, but mixed, effects on C stocks. The final foliage mass was increased from 3.4 g to 9.6 g due to the increase in C allocation to foliage (Figure 5G, light blue to green), which has a positive feedback on GPP. Concomitant changes in C allocation to wood and root resulted in smaller changes to these biomasses as shown in Figure 5H-I (2.5 g and 7.0 g rise respectively). Overall, the change in allocation pattern resulted in an increase in final total biomass by 15.74 g (Table 4). Growth respiration rate ($Y$) was ~20% lower in free seedlings (Figure 5D), which had a considerable impact on C budgets (the final total biomasses were increased by 9.56 g, Table 4 and green to yellow, Figure 5G-I). Leaf turnover, $s_f$ was low in the free seedlings compared to the 5 L container treatment (Figure 5E) which had a large positive effect on final C pools (Figure 5G-I, yellow to blue). The foliage mass was increased by 5.6 g; the wood and root masses were also further increased (3.4 g and 5.8 g respectively) due to the increase in GPP when foliage is retained for longer. Finally, the utilization coefficient, $k$ was higher in free seedlings (Figure 5F) causing a 20-30% positive feedback on C budgets (total biomass increased by 23.08 g, Table 4 and Figure 5G-I, blue to red).

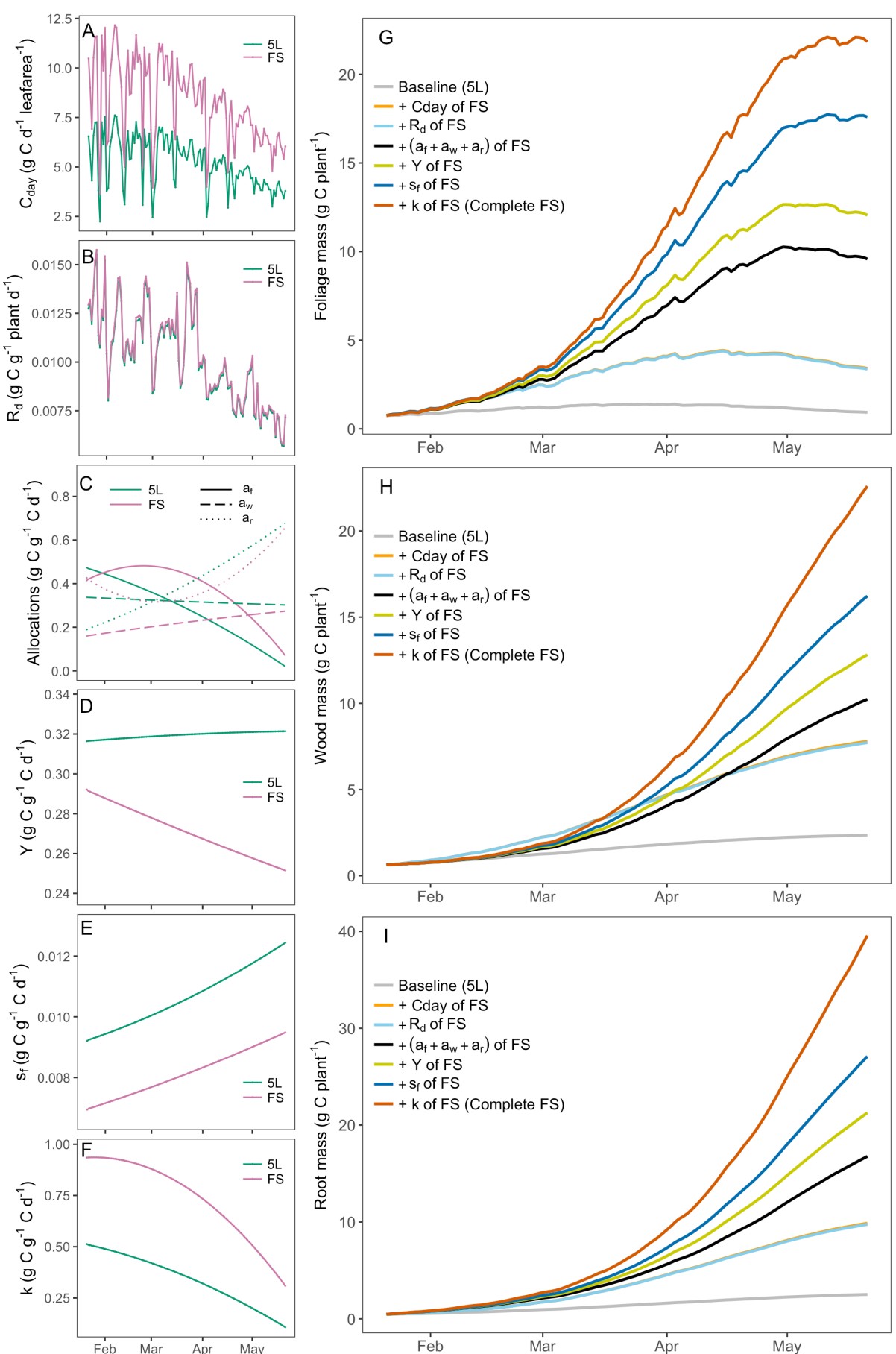

484

**Figure 5:** Attribution analysis. Left column (A-F): changes in inferred parameters; Right column (G-I): associated impacts on C budgets due to sequential parameter changes from 5 L container treatment to that of free seedling (right column, G-I). Different colours in the figure indicate the parameter shifts (left column, A-F) and their associated impacts on C budgets (right column, G-I). Legend: 5L, highly sink-limited treatment with container size of 5 L; FS, Free Seedling without any sink limitation. Note that the orange line is overlain by the light blue line: the small change in maintenance respiration results in a very minor effect on biomass growth.

**Table 4:** Estimates of final biomass due to parameter change (individual and sequential), showing the contribution of each parameter separately and successively to biomass changes. All values in g C plant$^{-1}$. +/- indicates biomass increase or decrease due to particular parameter change. The final column corresponds to the changes shown in Figure 5.

| Parameter change | Individually | | Sequentially |
|---|---|---|---|
| | 5 L » FS | FS » 5 L | 5 L » FS |
| Baseline $C_t$ | 5.81 | 83.99 | 5.81 |
| $C_{day}$ | +15.28 | -71.9 | +15.28 |
| $R_d$ | -0.08 | +1.1 | -0.24 |
| $(a_f + a_w + a_r)$ | +1.53 | -45.5 | +15.74 |
| Y | +0.41 | -19.22 | +9.56 |
| $s_f$ | +1.13 | -19.17 | +14.77 |
| k | +0.44 | -23.08 | +23.08 |
| | FS total observed $C_t$ | | 83.99 |

# 4 Discussion

## 4.1 Effects of sink limitation on C balance

Our DA-model analysis of this root volume restriction experiment provided significant new insights in the response of key C balance processes to sink limitation. We were able to infer that, in addition to a reduction in photosynthetic rates, sink limitation reduced NSC utilization rates, increased growth respiration, modified allocation patterns and enhanced senescence. Our attribution analysis indicated that all of these process responses contributed significantly to the overall reduction in biomass observed under low rooting volume.

We first tested the null hypothesis (H1) that seedling growth rates could be adequately predicted from current-day photosynthate. This hypothesis was rejected, with a storage pool being necessary to simulate growth, particularly for containerized seedlings (Sim A, Table 3). The approach of simulating growth from current-day photosynthate is commonly used in models, particularly for evergreen plants (e.g. (Jain and Yang, 2005; Law et al., 2006; Thornton et al., 2007)), but several authors have proposed the need for a storage pool to balance the C sources and sinks in the short term, as well as simulate the effects of photosynthetic down-regulation in the long-term (Pugh et al., 2016; Richardson et al., 2013; Fatichi et al., 2016). Our results support the need for an NSC pool in CBMs.

We then tested the second null hypothesis (H2) that there was no effect of treatment on the parameters of the C balance model. This hypothesis was also rejected: fitting the DA-model framework simultaneously to all treatments with one set of parameters (ignoring sink limitation effect) gave a low goodness of fit (Sim B, Table 1). This result is consistent with the finding of Campany et al. (2017) that the observed effects of sink limitation on photosynthesis in this experiment were insufficient to explain the large reduction in biomass. Instantaneous photosynthetic rates were reduced 20-30% by sink limitation. Our DA analysis indicated that several other processes contributed to the reduction in biomass growth, including carbohydrate utilization, growth respiration, allocation patterns, and turnover.

Our results suggested a significant effect of sink limitation on the carbohydrate utilization rate, $k$ (Figure 3A). The modelled $k$ values were approximately twice as large in free seedlings compared to the small containers. This result supports the hypothesis (H3) that plants would have the lowest utilization rate under sink-limited conditions. At the start of the measurement period, the free seedlings were utilizing almost all C produced immediately in growth ($k$ close to 1.0, Figure 3A). The utilization coefficient of the free seedlings decreased over time, causing a build-up in C storage (Figure 2D). This decrease in utilization rate could potentially be an ontogenetic effect, with free seedlings initially allocating all carbon to growth during establishment but increasing storage with increasing size. However, ontogenetic effects are confounded with season in this experiment, such that decreasing utilization rates over time could also be a result of decreasing temperatures moving into autumn. There is a real need to quantify how the carbohydrate utilization rate varies with environmental conditions and ontogeny; data assimilation of experiments in which photosynthesis and growth rates have been monitored over time offer one means to do so.

Although the carbohydrate utilization rate was highest in the free seedlings, leaf carbohydrate
concentrations were not lower in these plants at the end of the experiment. As shown in the
final C budget analysis (Figure 4), there was a higher total C allocation to the NSC pool in
free seedlings than sink-limited treatments. Final carbohydrate storage was high in free
seedlings despite high $k$ because the carbohydrate pool was recharged throughout the
experiment (Figure 2D), as the free seedlings had high photosynthetic rates but no higher
maintenance respiration requirement. In contrast, NSC was depleted for the smallest pot
treatments after mid-March (Figure 2D) when demand exceeded supply due to both limited
production of photoassimilates and enhanced leaf litterfall (Figure 3F).
The modelled rate for growth respiration, $Y$ was larger for sink limited treatments than the
free seedling (Figure 3B). Overall, there was lower C utilization (i.e. CUE) in plant structural
growth in sink-limited treatments (~45%) compared to free seedlings (~60%). This finding
supports the "wasteful plant" hypothesis H4.  Inferred $Y$ remained constant over time for the
containerized treatments, implying a fixed portion of C loss due to growth respiration despite
seasonal variation. However, a reduction in $Y$ over time was inferred for the free seedling,
suggesting a possible ontogenetic effect. However, it is important to note that we have
inferred growth respiration from the CBM framework. Therefore, these estimates could
possibly also include C losses via other pathways. Direct measurements of growth respiration
rates would be useful to confirm the inferred effects of sink limitation and investigate
potential underlying mechanisms.
We also demonstrated that the allocation fractions among organs change in sink-limited
conditions, with sizeable consequences for plant growth rates. Previous analyses of pot-size
experiments have generally only been able to estimate changes in final biomass partitioning
(e.g. Poorter et al. 2012a). Campany et al. (2017) analysed final biomass partitioning in the
experiment and did not find any significant difference in biomass partitioning in sink-limited
seedlings compared to free seedlings, once ontogenetic drift was taken into account. Our
analysis adds to that of Campany et al. (2017) by calculating the dynamics of allocation over
time and taking estimated foliage loss into account. Our analysis showed that modelled
allocation fractions vary significantly over time (Figure 3C, D and E). In the free seedlings,
allocation to foliage decreased, and allocation to both wood and roots increased, reflecting
the ontogenetic effects mentioned by Campany et al. (2017). However, our analysis also
highlights significant variations among treatments in the modelled C allocation fractions to
foliage, wood and root that are not ontogenetic. At the beginning of the experiment, foliage
allocation fractions were similar for all treatment groups, but wood allocation was higher, and
root allocation lower, in the containerized seedlings compared to the free seedlings. For the
containerized seedlings, changes over time also differed from those in the free seedlings:
wood allocation decreased marginally, rather than increasing, foliage allocation declined
steeply over time, and root allocation increased steeply. These allocation patterns in seedlings
supported our hypothesis H5 that sink limitation due to root restriction would favour
allocation to wood over foliage or fine roots. Calculating dynamic allocation patterns over the
course of an experiment thus provides additional insights beyond analysis of the final
biomass outcome.
**4.2    Application of DA to infer C balance processes**
We have demonstrated that the DA approach can be an invaluable tool for quantifying C
fluxes in experimental systems, enabling us to extract important new information from
existing datasets to inform carbon balance models, such as the rate and timing of the transfer
of photosynthate to and from storage pools. The DA-modelling approach is able to draw
together the experimental data to estimate all the components of C balance, including
photosynthesis, respiration, NSC, biomass partitioning and turnover. This approach could
readily be applied to other experiments to derive new information allowing better
representation of C balance processes in vegetation models.
Applying this approach requires a range of measurements to constrain the key C balance
processes. Here, we used estimated daily C assimilation and maintenance respiration rate as
model inputs and constrained the model with measurements of biomass pools (foliage, wood,
root) and foliage NSC concentrations. We used fortnightly foliage and wood biomass
measurements; the DA framework would work with fewer data observations, but parameters
would be estimated with less accuracy. Informal exploration of the model suggested that
measurements of foliage turnover would have been particularly useful to better constrain the
model. Any experiment having estimates of GPP, maintenance respiration, and structural
biomass could potentially be investigated with this framework. However, additional
measurements of storage and turnover would be highly beneficial for the performance of the
simulation. Repeated observations over time are also useful, particularly for young plants, to
account for variations in parameter values over time. We found significant changes in
parameter values during the course of the 4-month experiment, which may be linked to both
ontogeny and seasonal variation in temperature.
One major caveat on our results is that below-ground carbon cycling processes were not well
characterized. For practical reasons, processes such as root growth, respiration, turnover, and
exudation are rarely well quantified in empirical studies. Here, we had access to initial and
final estimates of root biomass. Root respiration was estimated; root turnover and exudation
were assumed to be zero. There is evidence that stress can increase rates of root exudation:
for example, Karst et al. (2016) demonstrate increased exudation rates in seedlings exposed
to cold soils. They also showed that stressed plants may exude C beyond that predicted by
simple concentration gradients in NSC between root and soil. The loss of C independent of
NSC in roots suggests that exudation may be actively enhanced once plant growth is limited
(Hamilton et al., 2008; Karst et al., 2017). As our CBM does not include this process, it
would attribute any C loss through root exudation to another process removing C from the
plant, such as growth respiration. The increase in growth respiration that we inferred may
thus potentially include root exudation. We have reasonable confidence, from the
combination of measurements available, in our inference that the C loss term was increased
with sink limitation. However, direct measurements of one or both processes would be
required to determine the role of root exudation.
In addition, we did not have access to estimates of root or wood NSC. We used data
measured in a previous experiment on 4-month old *E. globulus* seedlings (Duan et al. 2013)
to estimate these values from foliage NSC. It would have been useful to obtain these values,
particularly since wood and root tissue can act as storage organs, and the timing of storage
development would be extremely useful to quantify. The concentration of NSC in plant roots
measured by Duan et al. (2013) was relatively small compared to that of foliage (mean
2.15%). However, fine root NSC values in a nearby experiment on 17-month-old *E.*
*parramattensis* saplings were even lower (0.78%) (Morgan E. Furze et al. unpublished data).
It is possible that these very fast-growing Eucalypt species only start to accumulate root
reserves when they are established. Further research is needed to quantify the trade-off
between allocation to growth and storage during establishment.

## 4.3 Implications for modelling plant growth under sink limited conditions

The goal of our study was to examine how carbon balance models should be modified to represent sink limitation of growth, whilst maintaining mass balance. Our results demonstrate that several process representations need to be modified. Firstly, we demonstrate a clear need to incorporate a carbohydrate storage pool, with a dynamic utilization rate for growth. We demonstrate that the utilization rate is slowed by sink limitation, and may also vary with ontogeny. Targeted experimental work is needed to better quantify this variation in utilization rates. Secondly, in addition to a feedback on photosynthetic rates, other plant processes including growth respiration, turnover and allocation are also affected by sink limitation. Applying a DA-modelling framework to experimental data with rooting volume restriction has allowed us to quantify these effects in this experiment. Applying this approach more broadly would potentially allow us to identify general patterns that could then be formulated for inclusion into models.

The inferences on carbohydrate dynamics from seedling studies could be used to infer mature tree responses that can subsequently be integrated at ecosystem level and beyond using the concepts of Hartmann et al. (2018). We are enthusiastic to see the approach applied to other experiments, but there are likely to be gaps in the datasets to constrain the key C balance processes. Fortunately, the DA approach does not require continuous measurements of all of the C stocks and fluxes. In the absence of measurements, the model can be relied upon to project the time evolution of missing stocks and fluxes, although of course, the precision of model estimates and insights that can be gained, increases with data availability. DA can also be applied at ecosystem scale. There are several successful examples of DA being applied to forest growth, albeit without a focus on storage (e.g. Van Oijen (2008); Williams et al. (2005); Bloom et al. (2016); Quaife et al. (2008); Pinnington et al. (2016)). Overall, this approach provides important insights into the regulation of carbohydrate storage and would significantly advance our ability to predict the impacts of environmental changes on plant growth and vulnerability to stress.

**Data availability**

The raw data are freely available on Figshare (doi: https://doi.org/10.6084/m9.figshare.5125087.v3). The R source code to perform all the data processing and analysis to replicate the figures is freely available as a Git repository (https://github.com/kashifmahmud/DA_Sink_limited_experiment).


**Author contribution**

KM analyzed the data, developed the model code, performed the simulations and wrote the
paper. BEM conceived the idea and helped in data analysis. RAD and CC provided the
experimental data. BEM, RAD, CC and MGD provided in-depth editing of the manuscript.


**Competing interests**

The authors declare that they have no conflict of interest.


**Acknowledgements**

This research was supported by the Australian Research Council (Discovery, DP
DP160103436), the Hawkesbury Institute for the Environment, and Western Sydney
University. The authors wish to thank Burhan Amiji for his technical assistance and all
individuals from Hawkesbury Institute for the Environment who helped during the
experimental harvest. We thank Mathew Williams for advice on implementing the data
assimilation framework.

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
