# Peer review of "Inferring the effects of sink strength on plant carbon balance processes from experimental measurements"

_Biogeosciences, 2018_

## Referee Comment (RC1) · S. Fatichi (Referee) · 19 Mar 2018

**Overall Review**

The manuscript uses a data-assimilation technique to combine an essential but well thought model of carbon balance and plant growth with observations of photosynthesis, maintenance respiration, changes in biomass and NSC obtained in seedlings of Eucalyptos tereticornis planted in containers of different volumes and freely in the soil. The original experiment of Campany et al 2017 reproduced indeed different degrees of sink limitations. The data-assimilation technique allows the authors to infer time dynamics of model parameters (e.g., allocation fractions) and to quantify the relative
importance of different processes in downregulating plant-growth under sink limitations. The article shows that the reduction of photosynthesis rate due to sink limitation is not sufficient to explain the reduction of plant growth since other adjustments in NSC utilization, allocation dynamics, and modified respiration and leaf turnover rates are playing an important role. The inclusion of a NSC storage pool and the capability to account for sink limitations emerge as key model components if results of the experiment are to be reproduced. While the path toward modeling plant growth under sink-limitations in mature ecosystems and under various environmental conditions remain long, this contribution is surely an important advancement in the right direction. The article is very well written and presented and most important it is very novel with comparison to the existing literature. As far as I know, it is the first time evidence of carbon sink-limitations is presented so markedly and modeled in a realistic context. In summary, I am very positive concerning the content and conclusions of the article. I think the manuscript is making a very important contribution to the field and I sincerely congratulate the authors for this nice piece of work. In the following, I just have some minor comment that can be helpful to improve further the presentation of this work, especially the comments: P.16 Line 352-358 and P.19. Line 416.

Sincerely,

Simone Fatichi

Minor comments

P.2 Line 42. The reference "Bonan 2008" is not present in the reference list and if the authors refer to the article "Forests and climate change: forcings, feedbacks, and the climate benefits of forests. Science 2008, 320:1444–1449." I do not think it would be an appropriate reference here. I would rather search for something more related to "plant-growth and forest-growth modeling" rather than something related to Earth System Models.

P.2 Line 53 . I would suggest to add also Paul and Foyer 2001, very relevant here.

BGD
Paul MJ, Foyer CH. 2001. Sink regulation of photosynthesis. Journal of Experimental Botany 52: 1383–1400.

P.3. Line 67. It must be "Fatichi et al 2014".

P.5. Line 146. Maybe this is not a case that has been encountered in this article. However, how does the model work when NPP is negative and therefore maintenance respiration is larger than carbon assimilation? Is maintenance respiration generally subtracted by the non-structural storage or is it done for each of the tissue separately?

P.5 Line 171. The relative amount of NSC in the roots appear to be a very small number, 9% of the total, while generally one would expect a significant amount of non-structural C-storage in roots especially for seedlings and grasses. Do you have any explanation for this?

P.8. Line 202-206. How are you dealing with the heterogeneity in photosynthetic properties among leaves and among plants? Were they significant? I know that you wrote in Line 211, that you use the mean for each treatment; is this the mean of how many replicates? Did you average the photosynthetic and stomatal model parameters (Vc-max, g1, ...) or the A-Ci and stomatal conductance values?

P.9 Line 222-223. If I am reading correctly there are 18 (3x6) coefficient to determine for each treatment, maybe this can be written explicitly to compare with number of measurements (44 points) in line 231-233. This allows some redundancy even in the case of separating each container size.

P. 9. Line 243-247. Please explain better this part of the data assimilation methodology, as it is now it is not very clear to me.

P. 14. Line 319. There are not "bold values" in Table 3. Probably a formatting issue.

P.16 Line 352-358. The lowest utilization rate in seedling in small containers would theoretically lead to an accumulation of NSC, at least in relative terms, which is something we do not see in Figure 2 and Figure 4. The explanation for such non-intuitive results is Printer-friendly version

BGD
only provided in the discussion (Line 505-510) and justified as a temporal effect, where NSC first accumulates in seedling in small containers but then they are depleted by the higher respiration costs and leaf turnover rates. I think it would be quite interesting to see in Figure 2, NSC (Cn,f) reported as fraction of total C mass in foliage (Ct,f), e.g., Cn,f/Ct,f. This would serve the double purpose of explaining such a different dynamic in the use of NSC as the season progress and will provide the percentage of NSC in leaves that can be compared with other studies. This will likely highlight a higher concentration in seedling in small containers at the beginning of the season but a lower concentration at the end (as in Figure 4).

P. 18. Line 401-403. The authors for some reason never refer to the concept of Carbon Use Efficiency (CUE), but I suggest it would be useful here to explain the results. Substantially what they are saying is that CUE = (1 - (Rm,tot+Rg)/GPP) is higher in free seedling and it is reduced by sink limitations. Maybe, a figure showing the temporal evolution of CUE for the various treatments would be also an interesting piece of information.

P.19. Line 416. I am not sure if parameters were changed one at a time resetting the previous parameter to the original value or if effects are added up (which seems more the case from the presentation of results). If parameters are really changed "one at a time", this does not allow to reproduce all the interactions among parameters and can inflate the role of certain parameters. Therefore the total effect of Table 4 (54.8 gC plant-1) does not necessarily correspond with the real total effect, which is not reported for comparison. If the parameters are switched on sequentially then you will obtain the total effect but the importance of certain parameters will not be clearly separated, since it will depend on the adopted sequence of switchers. For instance, the role of "k" would be likely smaller when the interplay with the other parameters is considered. Now, I am not asking to running simulations with interactions among parameters since they would be an extremely high number (going factorial) and they will not add much to the overall discussion on the model results. However, this simplification and the specific
method needs to be stated explicitly and the difference in the total effect between the simulations changing parameters "one at a time" and the total observed effect needs to be mentioned, since it can provide an idea of the importance of the interaction among parameters.

P.20 Line 445-451. Table 4 and Figure 5. Following my previous comment, I wonder if it is not better to show the effect of each parameter independently rather than the sum of the effects of the parameters on the final biomass. I think it would be better to show the effect of each parameter by itself on the baseline rather than what is shown now. In any case, a clearer explanation of what is shown would be necessary.

P. 23. Line 471. I think with regards to the importance of the storage pool in models, Fatichi et al 2016 would fit well here. Fatichi S., C. Pappas, V. Y. Ivanov (2016). Modelling plant-water interactions: an ecohydrological overview from the cell to the global scale. WIREs Water, 3(3), 327-368, doi: 10.1002/wat2.1125

P.25. Line 550-552. Another way to say the same concept is that CUE is higher in free seedlings.

P. 26. Line 586. Karst et al 2016 is not in the reference list.

P. 27. Line 605-610. I agree with the authors, but there is still an important challenge of dealing with sink limitations in ecosystems encompassing tall-trees and heterogeneous vegetation types and for which observations for data-assimilation may not be available.

P. 29. Line 675-677. I thank the authors for referencing to my work, but this article is completely irrelevant for the current paper and indeed is just quoted by mistake.

BGD

---

## Referee Comment (RC2) · H. Hartmann (Referee) · 26 Mar 2018

**Overview**

The paper by Mahmud et al. presents a data assimilation exercise where data from a manipulative experiment on small trees had been 'assimilated' by a carbon balance model (CBM). The manipulation aimed to reduce root sink strength by constraining growth space of the root systems with varying pot sizes (5 – 30 L, in 5 L steps, 35L, and a 'free' treatment where trees were grown without limitation). On these trees biomass pools (structural biomass of leaves, wood and roots, non-structural carbohydrates, NSC, in leaves) were measured with different frequencies and used to constrain the CBM which simulated GPP based on parameters derived from punctual measurements of assimilation and respiration. Model runs with different structures (with/without a NSC 'storage' pool) were performed to test how important such a carbon buffer is for CBM simulations. The set of parameters of a best suited model (parametrized with three sink strength classes) was discussed with respect to plant carbon (allocation) dynamics in response to sink limitation. In addition, an attribution analysis was performed which aimed to provide information of the underlying mechanisms responsible for changes in biomass from sink limitation.

The authors highlight the need for including a 'storage' component in vegetation models and the usefulness of their approach for further investigations to 'develop appropriate representations of sink-limited growth in terrestrial biosphere models'.

**General comments**

This is a very nice project as it combines experimental manipulations with a data assimilation procedure. During the last years I have been running several experiments to manipulate the plant carbon balance. I have been thinking repeatedly about such a data assimilation approach to learn more about plant carbon dynamics and the underlying mechanisms. This study here does exactly this and I applaud the authors for making this progress.

That being said, I think that the interpretation of the data and the general presentation of the manuscript can be improved to increase its impact. For example, one of the main findings of the study, i.e. the importance of a storage component for (more) realistic simulations of plant functioning, is a strawman. Plants do have a storage component and of course models that specifically include carbon storage will be more realistic, in particular in situation where NSC may accumulate due to sink limitations.

In my opinion, the merits of the study are not the particular findings but rather the documentation of the potential of the data assimilation approach. The findings have to be taken with caution as the constraints from measurements are simply not sufficient to allow deeper insights into plant functioning. For example, measurements of assimilation and respiration have been done twice only and the photosynthetic parameters derived from these two measurements were used to estimate GPP over the whole season. How robust are these parameters for that purpose? Similarly, leaves were sampled every second week for NSC measurements and the structural biomass of stem and roots was determined only for the fourth months or at the end of the experiment, respectively. Given these limitations, how relevant are your inferences, for example, that sink limitation has led to reductions in photosynthetic rates or enhanced respiratory losses?

Additionally, NSC were measured in leaves only and their distribution among plant organs estimated with fixed parameters. For a study that specifically aims to highlight the role of NSC storage in plant functional processes, this is a critical shortcoming. Within the experimental period, there may have been substantial shifts in the NSC distribution across organs and this could have a substantial impact on the simulated carbon dynamics.

That being said, I think you should rather discuss the approach, its potential but also its limitations. One aspect, for example, is how well a study on seedlings can 'develop appropriate representations of sink-limited growth in terrestrial biosphere models'. Such models usually simulate mature trees, not seedlings. We have recently published a paper addressing this particular topic: how to make use of seedling studies for inferences on mature trees and modeling of vegetation dynamics (see reference from EEB below). Instead of too many inferences I would like to see a critical evaluation of your method, including an assessment of what data are needed to get better constraints for the model. I have done experiments with small trees in growth chambers where GPP and several components of the carbon balance have been assessed continuously or at high temporal resolution. Applying sink limitation (I used source limitation but also drought, which is also a form of sink limitation) in such an experimental setting would allow making much more robust inferences that with the data set used here. Hence, my suggestion is to move away from the current focus of interpreting plant functional responses and instead concentrate on presenting the approach as a promising avenue for how to gain insights into plant functioning.

I hope my comments can help increasing the paper's impact.

Henrik Hartmann

Specific comments:

Abstract: Please add what species you have been working with.

L 20: processes affected by growth? That doesn't make sense.

L 21: What do you mean by 'component processes'?

L27-29: Not much content in this sentence.

Introduction in general: The structure of several paragraphs is not ideal and reduces the logical flow. For example, on L 55 you start a paragraph by asking how to include source and sink limitations in models but then you move to NSC storage in models. I understand that storage allows buffering discrepancies in source and sink activity but this is not strictly related to the limitations.

A more logical flow would be to say that there is ongoing discussion about realistic implementations of NSC in vegetation models and that, because of their multiple roles in plant functioning, such an implementation also provides a buffer against discrepancies in source and sink activity.

L74-76: Quantify growth by manipulating rooting volume? That also does not make sense.

L 88: Very good point!

L90-97: See also paper by Klein T, Hoch G. 2015. Tree carbon allocation dynamics determined using a carbon mass balance approach. New Phytologist 205, 147-159.

L 105-123: The presentation of the hypothesis is very awkward. Could you present this please in a more accessible and appealing way? This is not a funding proposal but a text intended for keeping readers keen on reading on. Please rephrase and restructure to make this a flowing text.

L137-141: Relocate further up to L 129 (after Australia).

L 142: I suggest presenting the data first, then the model.

Table 1: Why is there no hypothesis for simulation set C?

Results: The results are presented in a very uncommon form. The text repeats the hypotheses (not useful) and reads more like a discussion than a presentation of results. I suggest adapting a more formal style so the reader knows to differentiate between results and interpretation of results.

L 329: You mean Fig. 2. Please correct figure numbering for the following figures also.

Figure 1 (actually Fig. 2): Add title to each panel (leaf, wood, root, NSC).

L 352: Shouldn't this section be presented before the modeling outcome? Parameters first, then the modeled pools?

L 413: Is this a sensitivity analysis?

L 418: This belongs into the methods section.

L 422: And this should go into the figure caption.

Table 4: Most of this information has been reported in Fig. 4 already.

L 460-465: The emphasis here is on inferences on processes that are poorly constrained. See my general comments.

L 466-467 (and at beginning of other paragraphs): Please avoid restating the hypotheses.

Discussion in general:

I'd relocate the focus to discuss the potential of the approach and move away from interpreting the model outcome with respect to plant functioning.

The discussion is somewhat lengthy and verbose. Please be more concise and to the point.

A few suggestions from my own work which are based on whole-plant assessments of the C balance:

**Hartmann H, Adams HD, Hammond WM, Hoch G, Landhäusser SM, Wiley E, Zaehle S**. 2018. Identifying differences in carbohydrate dynamics of seedlings and mature trees to improve carbon allocation in models for trees and forests. Environmental and Experimental Botany.

**Hartmann H, McDowell NG, Trumbore S**. 2015. Allocation to carbon storage pools in Norway spruce saplings under drought and low $CO_2$. Tree Physiology **35**, 243-252.

**Hartmann H, Trumbore S**. 2016. Understanding the roles of nonstructural carbohydrates in forest trees – from what we can measure to what we want to know. New Phytologist **211**, 386-403.

**Huang J, Hammerbacher A, Forkelova L, Hartmann H**. 2017. Release of resource constraints allows greater carbon allocation to secondary metabolites and storage in winter wheat. Plant Cell Environ **40**, 672-685.

---

## Author Comment (AC1) · 1 May 2018

Response to Referee #1

**Overall Review**

The manuscript uses a data-assimilation technique to combine an essential but well thought model of carbon balance and plant growth with observations of photosynthesis, maintenance respiration, changes in biomass and NSC obtained in seedlings of Eucalyptos tereticornis planted in containers of different volumes and freely in the soil. The original experiment of Campany et al 2017 reproduced indeed different degrees of sink limitations. The dataassimilation technique allows the authors to infer time dynamics of model parameters (e.g., allocation fractions) and to quantify the relative importance of different processes in downregulating plant-growth under sink limitations. The article shows that the reduction of photosynthesis rate due to sink limitation is not sufficient to explain the reduction of plant growth since other adjustments in NSC utilization, allocation dynamics, and modified respiration and leaf turnover rates are playing an important role. The inclusion of a NSC storage pool and the capability to account for sink limitations emerge as key model components if results of the experiment are to be reproduced. While the path toward modeling plant growth under sink-limitations in mature ecosystems and under various environmental conditions remain long, this contribution is surely an important advancement in the right direction. The article is very well written and presented and most important it is very novel with comparison to the existing literature. As far as I know, it is the first time evidence of carbon sink-limitations is presented so markedly and modeled in a realistic context. In summary, I am very positive concerning the content and conclusions of the article. I think the manuscript is making a very important contribution to the field and I sincerely congratulate the authors for this nice piece of work. In the following, I just have some minor comment that can be helpful to improve further the presentation of this work, especially the comments: P.16 Line 352-358 and P.19. Line 416.

We appreciate the reviewer's comments and careful reading of our manuscript.

**Minor comments**

P.2 Line 42. The reference "Bonan 2008" is not present in the reference list and if the authors refer to the article "Forests and climate change: forcings, feedbacks, and the climate benefits of forests. Science 2008, 320:1444–1449." I do not think it would be an appropriate reference here. I would rather search for something more related to "plant-growth and forest-growth modeling" rather than something related to Earth System Models.

We had intended to cite Bonan's book "Ecological Climatology" but in line with the reviewer's suggestion we have now chosen something more process-based about the modelling of forest growth: a review by Mäkelä et al. (2000).

P.2 Line 53. I would suggest to add also Paul and Foyer 2001, very relevant here.Paul MJ, Foyer CH. 2001. Sink regulation of photosynthesis. Journal of Experimental Botany 52: 1383–1400.

The reference will be added.

**P.3. Line 67. It must be "Fatichi et al 2014".**

This was a typo, will be corrected.

P.5. Line 146. Maybe this is not a case that has been encountered in this article. However, how does the model work when NPP is negative and therefore maintenance respiration is larger than carbon assimilation? Is maintenance respiration generally subtracted by the non-structural storage or is it done for each of the tissue separately?

We will modify the C balance model to clarify this situation of having negative NPP. The arrow representing the total maintenance respiration,  $R_{m,tot}$  will be relocated to connect with the non-structural C pool (Cn). So, the total daily inputs of GPP will directly enter into Cn pool and daily  $R_{m,tot}$  will be then subtracted from this pool before utilizing the reminder for growth.

P.5 Line 171. The relative amount of NSC in the roots appear to be a very small number, 9% of the total, while generally one would expect a significant amount of non-structural C-storage in roots especially for seedlings and grasses. Do you have any explanation for this?

There are not a lot of data on NSC of Eucalyptus seedling roots. We used data from an experiment with a related species, *Eucalyptus globulus*, grown in small pots until four months of age (Duan et al. 2013). There's no obvious reason why these plants would have exceptionally small root NSC contents. We have access to observations from *Eucalyptus parramattensis* saplings in another experiment, which show a higher root NSC fraction, but those plants were considerably bigger and older (~17 months). It is possible that these very fast-growing Eucalypts do not start to accumulate root reserves until they are well-established. We will add this discussion in the manuscript to explain the low root NSC.

P.8. Line 202-206. How are you dealing with the heterogeneity in photosynthetic properties among leaves and among plants? Were they significant? I know that you wrote in Line 211, that you use the mean for each treatment; is this the mean of how many replicates? Did you average the photosynthetic and stomatal model parameters (Vcmax,  $g1, \ldots$ ) or the A-Ci and stomatal conductance values?

Measurements of photosynthesis were made fortnightly throughout the experiment on one fully expanded leaf per plant (Campany et al. 2017). A-Ci curves were also measured twice during the experiment. Treatment effects on photosynthesis were detected immediately on newly produced (fully expanded) leaves and we did not observe variation over time in photosynthetic rates. Hence, the photosynthesis parameters were assumed not to change over time, but were specific for each treatment. This was also queried by reviewer 2. We will add text to clarify this point.

P.9 Line 222-223. If I am reading correctly there are 18 (3x6) coefficient to determine for each treatment, maybe this can be written explicitly to compare with number of

measurements (44 points) in line 231-233. This allows some redundancy even in the case of separating each container size.

The total counts of parameters and measurements will be added to the manuscript to show the actual number of redundancy. There are in fact 5 parameters to determine for each treatment group (as root allocation,  $a_r = 1 - a_f - a_w$ ), so a total of 15 (3x5) coefficients to determine for each treatment group, compared with total 44 data measurements available for each treatment.

**P. 9. Line 243-247. Please explain better this part of the data assimilation methodology, as it is not very clear to me.**

We will elaborate on this section to provide more in depth idea of the DA algorithm.

**P. 14. Line 319. There are not "bold values" in Table 3. Probably a formatting issue. Yes, it was a formatting issue, and will be corrected.**

P.16 Line 352-358. The lowest utilization rate in seedling in small containers would theoretically lead to an accumulation of NSC, at least in relative terms, which is something we do not see in Figure 2 and Figure 4. The explanation for such non-intuitive results is only provided in the discussion (Line 505-510) and justified as a temporal effect, where NSC first accumulates in seedling in small containers but then they are depleted by the higher respiration costs and leaf turnover rates. I think it would be quite interesting to see in Figure 2, NSC (Cn,f) reported as fraction of total C mass in foliage (Ct,f), e.g., Cn,f/Ct,f. This would serve the double purpose of explaining such a different dynamic in the use of NSC as the season progress and will provide the percentage of NSC in leaves that can be compared with other studies. This will likely highlight a higher concentration in seedling in small containers at the beginning of the season but a lower concentration at the end (as in Figure 4).

We will replace the NSC total plot with an NSC concentration plot in Figure 2, and move the NSC total plot to supplementary material.

P. 18. Line 401-403. The authors for some reason never refer to the concept of Carbon Use Efficiency (CUE), but I suggest it would be useful here to explain the results. Substantially what they are saying is that CUE = (1 - (Rm,tot+Rg)/GPP) is higher in free seedling and it is reduced by sink limitations. Maybe, a figure showing the temporal evolution of CUE for the various treatments would be also an interesting piece of information.

Carbon use efficiency (CUE), calculated as seedling biomass per unit total photosynthesis, was shown in Campany et al. (2017) and we had tried to avoid duplicating results that were already published. However, we will add numbers to this part of the results. The temporal evolution of CUE for various treatment groups is also interesting and will be added to the supplementary material.

P.19. Line 416. I am not sure if parameters were changed one at a time resetting the previous parameter to the original value or if effects are added up (which seems more the case from the presentation of results). If parameters are really changed "one at a time", this does not allow to reproduce all the interactions among parameters and can inflate the role of certain

parameters. Therefore the total effect of Table 4 (54.8 gC plant-1) does not necessarily correspond with the real total effect, which is not reported for comparison. If the parameters are switched on sequentially then you will obtain the total effect but the importance of certain parameters will not be clearly separated, since it will depend on the adopted sequence of switchers. For instance, the role of "k" would be likely smaller when the interplay with the other parameters is considered. Now, I am not asking to running simulations with interactions among parameters since they would be an extremely high number (going factorial) and they will not add much to the overall discussion on the model results. However, this simplification and the specific method needs to be stated explicitly and the difference in the total effect between the simulations changing parameters "one at a time" and the total observed effect needs to be mentioned, since it can provide an idea of the importance of the interaction among parameters.

The reviewer makes a good point. We suggest that we keep the attribution analysis shown in Figure 5 where the effects of various parameters are sequentially added up to get the total effect over the duration of the experiment, but that we modify Table 4 to present the effect of the parameters changing one at a time, resetting the previous parameter to the baseline value. This would fulfil both purposes, eventually showing the total biomass changes and the contribution of each individual parameter separately, along with the interaction among parameters. All these will be adjusted with explicit text to clarify the section.

P.20 Line 445-451. Table 4 and Figure 5. Following my previous comment, I wonder if it is not better to show the effect of each parameter independently rather than the sum of the effects of the parameters on the final biomass. I think it would be better to show the effect of each parameter by itself on the baseline rather than what is shown now. In any case, a clearer explanation of what is shown would be necessary.

According to the previous response, we will modify Table 4 to present the effect of the parameters changing one at a time resetting the previous parameter to the baseline value, which will illustrate the contribution of each individual parameter separately, along with the interaction among parameters.

P. 23. Line 471. I think with regards to the importance of the storage pool in models, Fatichi et al 2016 would fit well here. Fatichi S., C. Pappas, V. Y. Ivanov (2016). Modelling plantwater interactions: an ecohydrological overview from the cell to the global scale. WIREs Water, 3(3), 327-368, doi: 10.1002/wat2.1125

The reference will be added.

P.25. Line 550-552. Another way to say the same concept is that CUE is higher in free seedlings.

Yes, indeed. We will mention CUE while discussing the C utilization rate.

P. 26. Line 586. Karst et al 2016 is not in the reference list.

The reference will be added.

P. 27. Line 605-610. I agree with the authors, but there is still an important challenge of dealing with sink limitations in ecosystems encompassing tall-trees and heterogeneous vegetation types and for which observations for data-assimilation may not be available.

We agree with the reviewer that there are challenges to apply this approach at ecosystem scale due to data availability. However, the main focus of our paper is to build up a foundational step towards understand plant functioning rather than solving ecosystem problems. Moreover, one of the important features of DA is that it does not need all the data streams from every individual C stocks and fluxes. There are several successful examples of DA being applied to forest growth, albeit without a focus on storage (e.g. Bloom et al. (2016), Van Oijen (2008), Williams et al. (2005)). Since this was also a question raised by the second reviewer, we will add a short discussion on the potential for applying DA to investigate storage at ecosystem scale.

**P. 29. Line 675-677. I thank the authors for referencing to my work, but this article is completely irrelevant for the current paper and indeed is just quoted by mistake.**

This was a typo, and will be removed.

References:

Bloom, A.A., Exbrayat, J.-F., van der Velde, I.R., Feng, L. and Williams, M. (2016) The decadal state of the terrestrial carbon cycle: Global retrievals of terrestrial carbon allocation, pools, and residence times. Proceedings of the National Academy of Sciences 113(5), 1285-1290.

Campany, C.E., Medlyn, B.E. and Duursma, R.A. (2017) Reduced growth due to belowground sink limitation is not fully explained by reduced photosynthesis. Tree Physiol 37(8), 1042-1054.

Duan, H., Amthor, J.S., Duursma, R.A., O'Grady, A.P., Choat, B. and Tissue, D.T. (2013) Carbon dynamics of eucalypt seedlings exposed to progressive drought in elevated [CO2] and elevated temperature. Tree Physiology 33(8), 779-792.

Mäkelä, A., Landsberg, J., Ek, A.R., Burk, T.E., Ter-Mikaelian, M., Ågren, G.I., Oliver, C.D. and Puttonen, P. (2000) Process-based models for forest ecosystem management: current state of the art and challenges for practical implementation. Tree Physiology 20(5-6), 289-298.

Van Oijen, M. (2008) Bayesian Calibration (BC) and Bayesian Model Comparison (BMC) of Process-Based Models: Theory, Implementation and Guidelines.

Williams, M., Schwarz, P.A., Law, B.E., Irvine, J. and Kurpius, M.R. (2005) An improved analysis of forest carbon dynamics using data assimilation. Global Change Biology 11(1), 89-105.

---

## Author Comment (AC2) · 1 May 2018

Response to Referee #2

Overview
The paper by Mahmud et al. presents a data assimilation exercise where data from a manipulative experiment on small trees had been 'assimilated' by a carbon balance model (CBM). The manipulation aimed to reduce root sink strength by constraining growth space of the root systems with varying pot sizes (5 – 30 L, in 5 L steps, 35L, and a 'free' treatment where trees were grown without limitation). On these trees biomass pools (structural biomass of leaves, wood and roots, non-structural carbohydrates, NSC, in leaves) were measured with different frequencies and used to constrain the CBM which simulated GPP based on parameters derived from punctual measurements of assimilation and respiration. Model runs with different structures (with/without a NSC 'storage' pool) were performed to test how important such a carbon buffer is for CBM simulations. The set of parameters of a best suited model (parametrized with three sink strength classes) was discussed with respect to plant carbon (allocation) dynamics in response to sink limitation. In addition, an attribution analysis was performed which aimed to provide information of the underlying mechanisms responsible for changes in biomass from sink limitation. The authors highlight the need for including a 'storage' component in vegetation models and the usefulness of their approach for further investigations to 'develop appropriate representations of sink-limited growth in terrestrial biosphere models'.

We appreciate the reviewer's comments and careful reading of our manuscript.

General comments
This is a very nice project as it combines experimental manipulations with a data assimilation procedure. During the last years I have been running several experiments to manipulate the plant carbon balance. I have been thinking repeatedly about such a data assimilation approach to learn more about plant carbon dynamics and the underlying mechanisms. This study here does exactly this and I applaud the authors for making this progress.

We appreciate the reviewer's positive comments, and are hopeful that he will find our approach useful in future. To aid uptake of our approach, we have made all code freely available as a Git repository:
https://github.com/kashifmahmud/DA_Sink_limited_experiment

That being said, I think that the interpretation of the data and the general presentation of the manuscript can be improved to increase its impact. For example, one of the main findings of the study, i.e. the importance of a storage component for (more) realistic simulations of plant functioning, is a strawman. Plants do have a storage component and of course models that specifically include carbon storage will be more realistic, in particular in situation where NSC may accumulate due to sink limitations.

Here we must respectfully disagree with the reviewer. Although in reality plants do have a storage component, it is not necessarily the case that including such a storage

component in the model will lead to model improvements. Making a model more and more complex to "better represent reality" is unwarranted in many situations – the idea of a model is to abstract reality, not reproduce it. Hence, it is actually quite important to test whether or not adding the storage component improves the performance of the model enough to justify the additional complexity. To address this comment, we will add a sentence to the introduction to explain why we feel it necessary to test the "strawman" null hypothesis.

In my opinion, the merits of the study are not the particular findings but rather the documentation of the potential of the data assimilation approach. The findings have to be taken with caution as the constraints from measurements are simply not sufficient to allow deeper insights into plant functioning. For example, measurements of assimilation and respiration have been done twice only and the photosynthetic parameters derived from these two measurements were used to estimate GPP over the whole season. How robust are these parameters for that purpose? Similarly, leaves were sampled every second week for NSC measurements and the structural biomass of stem and roots was determined only for the fourth months or at the end of the experiment, respectively. Given these limitations, how relevant are your inferences, for example, that sink limitation has led to reductions in photosynthetic rates or enhanced respiratory losses?

We are grateful that the reviewer recognises the potential of the DA approach to provide insight into C balance processes. However, we must also respectfully disagree with his comment that the measurements "are simply not sufficient to allow deeper insights into plant functioning". Although some measurements we'd like to have had are missing – notably root and stem NSC, growth respiration, turnover, and root exudation – we believe that this particular experiment is sufficiently well-constrained to draw the inferences that we have.

The reviewer mentions photosynthesis. Although we only used two sets of A-Ci data to estimate the parameters, net leaf photosynthesis was measured fortnightly throughout the experiment (Campany et al. 2017) and we can be confident from the lack of trend over time that these measurements are robust and representative. We will add information on this to the methods.

The height, diameter and leaf area were measured fortnightly throughout the experiment, and root biomass is also constrained at the end. As we note in the discussion, there is some question over whether our "Y" value represents growth respiration or other carbon losses from the plant, such as root exudation, but we have reasonable confidence, from the combination of measurements available, in our inference that the C loss term is increased with sink limitation.

No experiment can perfectly quantify the C balance – there will always be data gaps. We are enthusiastic to see our approach applied to other experiments, such as those described by the reviewer, but there are likely also to be gaps in these datasets to constrain the key C balance processes. Fortunately, the DA approach does not need to

have all of the data streams from each of the C stocks or fluxes, and can estimate some missing C stocks from other measurements, although of course the precision of model predictions increases with data availability. We already covered the impact of missing data streams at some length in the discussion; we will refine this discussion to more directly address the reviewer's comment.

Additionally, NSC were measured in leaves only and their distribution among plant organs estimated with fixed parameters. For a study that specifically aims to highlight the role of NSC storage in plant functional processes, this is a critical shortcoming. Within the experimental period, there may have been substantial shifts in the NSC distribution across organs and this could have a substantial impact on the simulated carbon dynamics.

The sink-limited container experiment only measured foliage NSC, and therefore to estimate the partitioning of the non-structural C among different organs, we used data from a different experiment on similar-sized seedlings of a related species (*Eucalyptus globulus*) (Duan et al. 2013). In that experiment, root, stem and wood NSC were measured repeatedly over two months and there was no statistically significant difference in the NSC distribution. We thus believe this is a justifiable assumption. We will add this information to the discussion, and acknowledge this as an uncertainty.

That being said, I think you should rather discuss the approach, its potential but also its limitations. One aspect, for example, is how well a study on seedlings can 'develop appropriate representations of sink-limited growth in terrestrial biosphere models'. Such models usually simulate mature trees, not seedlings. We have recently published a paper addressing this particular topic: how to make use of seedling studies for inferences on mature trees and modeling of vegetation dynamics (see reference from EEB below). Instead of too many inferences I would like to see a critical evaluation of your method, including an assessment of what data are needed to get better constraints for the model. I have done experiments with small trees in growth chambers where GPP and several components of the carbon balance have been assessed continuously or at high temporal resolution. Applying sink limitation (I used source limitation but also drought, which is also a form of sink limitation) in such an experimental setting would allow making much more robust inferences that with the data set used here. Hence, my suggestion is to move away from the current focus of interpreting plant functional responses and instead concentrate on presenting the approach as a promising avenue for how to gain insights into plant functioning. I hope my comments can help increasing the paper's impact.

We will add a discussion of the implications of this study for larger plants, drawing on the review paper mentioned by the reviewer. We already have a section where we discuss potential and limitations of this approach and also identify the data needed to better constrain the model framework in article 4.2 (line 567-580). We will refer to the papers suggested by the reviewer as examples of experiments where this approach could be useful. However, we are reluctant to remove our focus on drawing inferences from the experimental study here. As described above, we believe that our inferences are valid - to the extent that we already describe in the Discussion - and help to

demonstrate the utility of the approach. Hence, we respectfully suggest that we should continue to do both: interpret the responses and present the approach.

Specific comments:
Abstract: Please add what species you have been working with.
> Will be added.

L 20: processes affected by growth? That doesn't make sense.
> The clause read "processes affected by growth under sink limitation". We will shorten to "processes affected by sink limitation"

L 21: What do you mean by 'component processes'?
> We meant various C cycle processes contributing to growth e.g. photosynthesis, respiratory losses, utilization of NSC, allocation pattern, turnover rates as mentioned in the following sentence. There is not space in the abstract to further expand on this; we will elaborate at line 82 in the Introduction.

L27-29: Not much content in this sentence.
> We will restructure the sentence.

Introduction in general: The structure of several paragraphs is not ideal and reduces the logical flow. For example, on L 55 you start a paragraph by asking how to include source and sink limitations in models but then you move to NSC storage in models. I understand that storage allows buffering discrepancies in source and sink activity but this is not strictly related to the limitations. A more logical flow would be to say that there is ongoing discussion about realistic implementations of NSC in vegetation models and that, because of their multiple roles in plant functioning, such an implementation also provides a buffer against discrepancies in source and sink activity.
> We will reorganize the paragraph with the addition of referee's observation on ongoing discussion about the realistic implementations of NSC in vegetation models.

L74-76: Quantify growth by manipulating rooting volume? That also does not make sense.
> We will omit "to quantify growth" from this sentence.

L 88: Very good point!
> We are not quite sure which point the reviewer is referring to here. However, as it is positive, no revision is needed.

L90-97: See also paper by Klein T, Hoch G. 2015. Tree carbon allocation dynamics determined using a carbon mass balance approach. New Phytologist 205, 147-159.
> We will add a reference to this paper as it's very relevant.

L 105-123: The presentation of the hypothesis is very awkward. Could you present this please in a more accessible and appealing way? This is not a funding proposal but a text

intended for keeping readers keen on reading on. Please rephrase and restructure to make this a flowing text.

The presentation of hypotheses is evidently a matter of taste. We felt that, given we had several different hypotheses to test, it made sense to lay them all out clearly and concisely in this way. We ask for the editor's advice here: we can turn this part into regular paragraphs if he feels it will improve the communication of our ideas.

L137-141: Relocate further up to L 129 (after Australia).

Will be relocated.

L 142: I suggest presenting the data first, then the model.

We will swap the subsection 2.2 and 2.3 to follow reviewer's comment.

Table 1: Why is there no hypothesis for simulation set C?

Simulation set C aimed to quantify the effect of changes individual parameters on overall seedling growth, and that is why this section does not come with any hypothesis.

Results: The results are presented in a very uncommon form. The text repeats the hypotheses (not useful) and reads more like a discussion than a presentation of results. I suggest adapting a more formal style so the reader knows to differentiate between results and interpretation of results.

Again, we refer to the editor for advice. It is true that for many papers, it is appropriate to present the results without reference to the hypotheses, and only return to the hypotheses in the discussion. However, here we felt it was necessary to address the hypotheses directly in the results to better guide the reader through the outcomes of the different model analyses. For example, consider section 3.1, where we present the BIC values for different simulation sets. We could here reduce the paragraph to just lines 299-303:

"Table 3 (Simulation Set A) shows the results for model fits with the optimal grouping strategy (three treatment groups). BIC values were consistently lower for the model including the storage pool; the improvement in model fit is most noticeable for the containerized seedlings."

and move the sentences describing how the hypothesis is tested to the discussion. Our personal feeling is that this makes this result paragraph fairly impenetrable. However, we are happy to take the editor's decision on this.

L 329: You mean Fig. 2. Please correct figure numbering for the following figures also.

We have checked the Figure numbers on the pdf version from BGD library and believe that all figure numbers are correct.

Figure 1 (actually Fig. 2): Add title to each panel (leaf, wood, root, NSC).

We will add the titles to each panel.

 Shouldn't this section be presented before the modeling outcome? Parameters first, then the modeled pools?

When writing the manuscript, we experimented with both orders of presentation. We found this order of presentation to be more logical. Showing the data first allows the reader to see how the DA works in terms of data fit, before moving on to examine the model parameters.

L 413: Is this a sensitivity analysis?

No, it's something a bit different from a sensitivity analysis, which is why we call it an attribution analysis. A sensitivity analysis aims to quantify the sensitivity of model outputs to a change in one or more input parameters. What we are doing is trying to work out (attribute) how much of the observed change in growth is due to the observed change in each parameter.

L 418: This belongs into the methods section.

We have already mentioned the procedure in methods; this sentence was intended to remind the readers that the analysis attributes the change in biomass from the smallest container to free seedling. We will omit this sentence.

L 422: And this should go into the figure caption.

Will be moved to figure caption.

Table 4: Most of this information has been reported in Fig. 4 already.

According to the response to reviewer 1, we will modify Table 4 to present the effect of the parameters changing one at a time resetting the previous parameter to the baseline value, which will illustrate the contribution of each individual parameter separately, along with the interaction among parameters.

L 460-465: The emphasis here is on inferences on processes that are poorly constrained. See my general comments.

The beauty of the DA approach is that it is possible to make inferences about processes that are individually poorly constrained, but contribute to overall growth, and hence can be constrained against total growth. Here, as mentioned above, we have 15 coefficients to determine against 44 mean data values, each of which is determined by 7 replicates. This total amount of data enables us to fully constrain the model. We will add this explanation to the discussion.

L 466-467 (and at beginning of other paragraphs): Please avoid restating the hypotheses.

Again, this is a matter of taste, and we will defer to the editor. The reviewer's point is that we have already stated the hypotheses in the introduction, so there is no need to re-state them here. Our preference is to re-state them, otherwise the reader will need to

keep flipping back to the introduction to check which one "H2" was again, and we think that could be annoying.

Discussion in general:
I'd relocate the focus to discuss the potential of the approach and move away from interpreting the model outcome with respect to plant functioning. The discussion is somewhat lengthy and verbose. Please be more concise and to the point.

As explained above, we prefer to keep both emphases in the discussion. We will go through the discussion and attempt to reduce the length and verbosity.

A few suggestions from my own work which are based on whole-plant assessments of the C balance:
Hartmann H, Adams HD, Hammond WM, Hoch G, Landhäusser SM, Wiley E, Zaehle S. 2018. Identifying differences in carbohydrate dynamics of seedlings and mature trees to improve carbon allocation in models for trees and forests. Environmental and Experimental Botany.
Hartmann H, McDowell NG, Trumbore S. 2015. Allocation to carbon storage pools in Norway spruce saplings under drought and low CO2. Tree Physiology 35, 243-252.
Hartmann H, Trumbore S. 2016. Understanding the roles of nonstructural carbohydrates in forest trees – from what we can measure to what we want to know. New Phytologist 211, 386-403.
Huang J, Hammerbacher A, Forkelova L, Hartmann H. 2017. Release of resource constraints allows greater carbon allocation to secondary metabolites and storage in winter wheat. Plant Cell Environ 40, 672-685.

We have read through the suggested papers and will cite appropriately where relevant.

References:
Campany, C.E., Medlyn, B.E. and Duursma, R.A. (2017) Reduced growth due to belowground sink limitation is not fully explained by reduced photosynthesis. Tree Physiol 37(8), 1042-1054.
Duan, H., Amthor, J.S., Duursma, R.A., O'Grady, A.P., Choat, B. and Tissue, D.T. (2013) Carbon dynamics of eucalypt seedlings exposed to progressive drought in elevated [CO2] and elevated temperature. Tree Physiology 33(8), 779-792.

---

## Author Response (AR2)

Associate Editor Decision: Publish subject to minor revisions (review by editor) (04 Jun 2018) by Sönke Zaehle

**Response to Associate Editor:**

Overall Review

Many thanks for your revised manuscript, which does satisfactorily address all concerns raised by the reviewers. Based on my own reading of the revised manuscript, I have a small range of minor, editorial suggestions to help improve the clarity of the manuscript. Looking forward to receiving a final version of the manuscript in short time.

We appreciate the Associate Editor's comments and careful reading of our manuscript.

**Minor editorial comments:**

L 56: is the "and" necessary here? In my view the implementation in models is solely discussed because of the multiple roles NCSs are assumed to play in plants.

The "and" is now removed (line 56).

L110-113. I would avoid mixing hypotheses and the justification of your approach here. Maybe having these added sentences after the added sentence in L107, and then continuing "Therefore, we tested ... " would be clearer?

The sentences are reorganised as suggested (line 110-116).

L209: Am I to deduce from this that you assume that the tissue-specific dark respiration rate of leaves rate of used to estimate maintenance respiration rates also for woody and root tissues? What's the justification for this simplification?

This assumption is based on work by Drake et al. (2017) showing tissue-specific dark respiration rates of different organs are similar in *Eucalyptus tereticornis* seedlings. We have added this reference in the text (line 206-209).

eq 7: Does not use a\_r. This makes it inconsistent with the text in 223-227 and the figure 1. Why not use a\_r in the equation, but add to caption and L223-227 that a\_r is defined as 1-af-aw. This would avoid introducing an inline equation (which should be avoided) in line 247

All these are adjusted according to the comment (eq 7, line 226, 233, 254).

L231 (eq 8-10): Probably hair-splitting, but actually these equations should read  $Ct, f = k_n, f * C_n + C_s, f$ , etc

for consistency with eq 4-7 and you probably should repeat the partitioning now mentioned in L178.

The equations are modified according to the suggestion (eq 8-10, line 235).

L443: I think that the use of the term "one at a time" is misleading here, because this term strictly means keeping all other parameters at their standard values (corresponding to your individual scheme). From reading this text, I would assume that Figure 5 shows exactly what happens if each parameter is change once to "free" while all the others are kept at 5L, which is not what you've done nor what figure 5 shows.

The confusion is now clarified in line 451.

L444-5: I wonder if this would be clearer if you stated explicitly that you change from the parameter set derived from DA on the 5L observations to that of the parameters obtained when using the free seedlings as constraint of the model? Please design and label table 4 such that it is not necessary to refer to "columns" of the Table in the text.

The text (line 459-462) and the table are refined.

L453: This sentence reads repetitive from the preceding ones. Please make sure that the newly added text is better integrated

The paragraph is reorganised and the problem of repetition is now resolved.

L448 remove (+/-)

Removed from line 454.

L448: rather remove "both", and add ", respectively" in the end? Modified (line 454-455).

L563: add "in seedlings". Added in line 575.

Figure 3 caption. remind the reader that ar is implied from af and aw, for instance by putting this explicit in the Yaxis label. Simply to avoid the reader assuming you have 6 free parameters-

Figure 3 caption and Y-axis label are modified.

Table 1: Be more explicit to state "leaf area feedback on photosynthesis and Rm"? Stated explicitly.

Table 1: Simset C better "5L and free seedlings treatment considered" Altered.

Table 1: I am not sure that "parameters changes one at a time" describes what you do here when you change an increasing number of parameters from the DA result of 5L to free Corrected.

Figure 4 caption: Add "Simulated" (or similar) to the beginning of the first sentence. Assuming this is what is shown. Remove duplicate bracket in "Container size (L))" Amended. Figure 5 caption: I am unsure about the term "input" parameters, as these plots describe the prescribed change in model parameters, but they are not really the "input", which are temperature and ?

Changed to "inferred" parameters.

Figure 5 caption: Please reminder the reader that you are sequentially changing the parameter values from 5L to free.

Repeated the simulation scenario in figure caption.

Figure 5 caption: I think this figure would be much easier to follow if panels A-F used always the same colours for 5L and free. The link between the colours in A-F and G-I is not evident, certainly not for daltonians.

The colours of the panels A-F are now adapted having the same colours for 5 L and free seedlings in all 6 panels.

SI: Unclear what "optimum" parameter settings are. Do you mean the DA posterior? Yes, indeed. The text is now clarified in S1 figure caption.

[revised manuscript text omitted]
</li> <li>Three treatment groups</li> <li>Not constrained with NSC data</li> <li>No leaf area feedback</li>                                                            | HI                    |
| В                 | Identify effect of
sink limitation on
model parameters                       |  <li>DA applied to estimate parameters for
model with storage pool</li> <li>Data divided into one, two, three or seven
treatment groups</li> <li>Constrained with NSC data</li> <li>No leaf area feedback</li>                                                          | H2-H5                 |
| С                 | Attribute overall
effect on growth
to changes in
individual
parameters |  <li>Forward model runs to quantify impact of individual processes on overall plant growth</li> <li>5L and free seedlings treatments considered</li> <li>Parameters changed individually and sequentially</li> <li>Leaf area feedback on photosynthesis and Rm</li>  |                       |

Table 2: Prior parameter PDFs (with uniform distribution) and the starting point of theiteration for all parameters

| Parameter                                                             | Minimum | Maximum | Starting value |
|-----------------------------------------------------------------------|---------|---------|----------------|
| k                                                                     | 0       | 1       | 0.5            |
| Y                                                                     | 0.2     | 0.4     | 0.3            |
| $a_{\mathrm{f}}$                                                      | 0       | 1       | 0.5            |
| $a_{ m w}$                                                            | 0       | 1       | 0.5            |
| Sf                                                                    | 0       | 0.01    | 0.005          |
| $a_{\rm r} = 1 - (a_{\rm f} + a_{\rm w})$ , where $0 < a_{\rm r} < 1$ |         |         |                |

**281 2.4.1 Importance of storage pool**

[revised manuscript text omitted]

**344 **3.2** Sink limitation effect on C balance processes**

345 We addressed our second null hypothesis (H2), that there is no effect of sink limitation on carbon balance processes, by comparing BIC values obtained for model fits when all 346 347 treatments were combined vs separating the treatments into sub-groups. If there was no effect 348 of sink limitation, the BIC value when all treatments are fit together would be similar to that 349 obtained when treatments are separated into groups. The BIC values shown in Table 3 350 (Simulation Set B) decrease strongly as number of treatment groups increases, indicating a 351 clear effect of sink limitation on carbon balance processes. Although the BIC values continue to decrease as more treatment groups are considered, we also found that interpreting 352 353 parameter changes became more difficult as the number of groups increased. Hence, further 354 analyses in this paper used unique parameter sets for three treatment groups: small containers, large containers, and free seedlings. 355

**356 3.3 Analysis of carbon stock dynamics**

357 Figure 2 shows the correspondence between modeled C pools and data. The model reproduced the key features of biomass growth over time in response to treatment. Biomass 358 growth (Figure 2A, B and C) and the foliage storage pool (Figure 2D) were very clearly 359 impacted by sink limitation: biomass growth was strongly reduced for containerised 360 361 seedlings, which was very well mimicked by the model. Foliage growth in the free seedlings 362 slowed towards the end of the experiment. Wood and root growth continued throughout the 363 experiment in freely-rooted seedlings but slowed down during the second half of the 364 experiment in containerized seedlings. NSC concentrations  $(C_{n,f} / C_{t,f})$  in seedlings in small 365 containers were higher compared those in free seedlings at the beginning of the season but all treatments had similar concentrations after four months (Figure 2D). In March, at the time of 366 367 the first leaf NSC measurements, the foliage storage pool (Supplementary Figure S1) was similar in size across all treatments, but it increased over time in the free seedlings as these 368 369 plants continued to grow, and decreased over time in the plants in small containers.

Modelled C stocks for all 7 treatments closely tracked their corresponding observations (Figure 2) as most of the predicted biomass values were within one standard error of the measurements. The exception is the 35 L container treatment, which is underestimated slightly because the grouping of 20, 25 and 35 L treatments into one group makes it difficult for the model to fit all treatments in this group.